# LaCl₃-based sodium halide solid electrolytes with high ionic conductivity for all-solid-state batteries

Chengyu Fu[1,12], Yifan Li[2,12], Wenjie Xu[3,4,12], Xuyong Feng [1,5] ✉, Weijian Gu[1], Jue Liu [6], Wenwen Deng[7], Wei Wang[8], A. M. Milinda Abeykoon[9], Laisuo Su [10], Lingyun Zhu [11], Xiaojun Wu [2] & Hongfa Xiang [1,5] ✉

To enable high performance of all solid-state batteries, a catholyte should demonstrate high ionic conductivity, good compressibility and oxidative stability. Here, a LaCl₃-based Na⁺ superionic conductor ($Na_{1-x}Zr_xLa_{1-x}Cl_4$) with high ionic conductivity of $2.9 \times 10^{-4}\,S\,cm^{-1}$ (30 °C), good compressibility and high oxidative potential (3.80 V $vs.$ Na₂Sn) is prepared via solid state reaction combining mechanochemical method. X-ray diffraction reveals a hexagonal structure ($P6_3/m$) of $Na_{1-x}Zr_xLa_{1-x}Cl_4$, with Na⁺ ions forming a one-dimensional diffusion channel along the $c$-axis. First-principle calculations combining with X-ray absorption fine structure characterization etc. reveal that the ionic conductivity of $Na_{1-x}Zr_xLa_{1-x}Cl_4$ is mainly determined by the size of Na⁺-channels and the Na⁺/La³⁺ mixing in the one-dimensional diffusion channels. When applied as a catholyte, the NaCrO₂||Na₀.₇Zr₀.₃La₀.₇Cl₄||Na₃PS₄||Na₂Sn all-solid-state batteries demonstrate an initial capacity of 114 mA h g⁻¹ and 88% retention after 70 cycles at 0.3 C. In addition, a high capacity of 94 mA h g⁻¹ can be maintained at 1 C current density.

Rechargeable sodium-ion all-solid-state batteries (ASSBs) have attracted much attention in recent years because of their safety and cost advantages over conventional lithium-ion batteries with liquid electrolytes[1–4]. Sodium solid-state electrolytes (SSEs) are one of the critical components in sodium-ion ASSBs, dictating the electrochemical performance of ASSBs. To ensure high power density, high-energy density, long cycle life, and low cost of ASSB, SSE should exhibit high ionic conductivity in a wide temperature range, broad electrochemical window, excellent compatibility with electrode materials, and low cost. Various types of sodium SSEs have been developed for their applications in sodium ASSBs, including organic polymers[1], inorganic sulfides[5–12], oxides[13–16], halides[4,17–23], and borohydrides[24–26].

Polymer-based SSEs are cheap and easy to process, but the ionic conductivity cannot match the industry requirement[1]. Oxides demonstrate ionic conductivity up to $10^{-3}\,S\,cm^{-1}$, but they are incompatible with cathode materials (e.g., Na₃V₂(PO₄)₃), leading to enormous interface resistance between the SSE and the cathode[13,14,27]. Sulfides show the highest ionic conductivity of $10^{-2}\,S\,cm^{-1}$, but its

¹School of Materials Science and Engineering, Hefei University of Technology, Hefei 230009 Anhui, China. ²School of Chemistry and Material Sciences, University of Science and Technology of China, Hefei, Anhui 230026, China. ³National Synchrotron Radiation Laboratory, University of Science and Technology of China, Hefei, Anhui 230029, China. ⁴School of Nuclear Science and Technology, University of Science and Technology of China, Hefei, Anhui 230029, China. ⁵Engineering Research Center of High-Performance Copper Alloy Materials and Processing, Ministry of Education, Hefei University of Technology, Hefei 230009 Anhui, China. ⁶Neutron Scattering Division, Oak Ridge National Laboratory, Oak Ridge, TN 37831, USA. ⁷Materials Science and Devices Institute, Suzhou University of Science and Technology, Suzhou, Jiangsu 215009, China. ⁸CAS Key Laboratory of Design and Assembly of Functional Nanostructures, Fuzhou 360002, China. ⁹Brookhaven National Laboratory, National Synchrotron Light Source II, Upton, New York, NY, USA. ¹⁰Department of Materials Science and Engineering, University of Texas at Dallas, Richardson, TX, USA. ¹¹School of Materials Science and Engineering, Anhui University Hefei 230601, China. ¹²These authors contributed equally: Chengyu Fu, Yifan Li, Wenjie Xu. ✉e-mail: 2021800026@hfut.edu.cn; hfxiang@hfut.edu.cn

inferior electrochemical stability results in poor cycle stability when coupling with Na metal anodes and layered structure cathodes[3,12,28]. Borohydrides also have high ionic conductivity, but the low oxidation potential is unstable against high-potential cathodes[24,26]. In comparison, halides have been considered as a promising SSE because of their high ionic conductivity, great deformability, adequate chemical stability, and good oxidative stability against cathodes. For example, plenty of lithium-ion halide SSEs with high ionic conductivity up to $10^{-2}$ S cm$^{-1}$ have been developed, such as $Li_3YCl_6$, $Li_3InCl_6$, $Li_3ScCl_6$, $Li_2ZrCl_6$, and $LiTaOCl_4$ et al.[29–35].

Several sodium halide SSEs have also been developed recently, such as $Na_2ZrCl_6$, $Na_3YCl_6$, $Na_3ErCl_6$, $NaAlCl_4$, and their derivatives[4,18,21–23]. Although these sodium halides exhibit oxidation stability as good as lithium halides, their sodium ionic conductivity is much lower (~$10^{-5}$ S cm$^{-1}$ at room temperature). Thus, it is urgent to develop sodium halide SSEs with high ionic conductivity. Typical halide electrolyte is composed of $MX_6$ (M = Y$^{3+}$, Zr$^{4+}$, Sc$^{3+}$ et al. cations, X = F$^-$, Cl$^-$, Br$^-$, I$^-$ et al. halides) octahedron frame and alkali metal ions (Li$^+$, Na$^+$, et al.) are distributed in the interstitial sites. Compared to lithium SSEs, the charge carrier of Na$^+$ in sodium SSEs is much larger than Li$^+$ (106 pm vs. 76 pm), which requires a significantly larger diffusion channel to ensure fast Na$^+$ ion diffusion.

Broaden the ion diffusion channel by adopting anions or cations with larger sizes is a promising strategy to increase the ionic conductivity of SSEs. For example, NASICON is a typical SSE structure, which can achieve high Li$^+$/Na$^+$ ion conductivity up to $10^{-3}$ S cm$^{-1}$ with optimized compositions, such as $Li_{1+x}Al_xGe_{2-x}(PO_4)_3$, $Li_{1+x}Al_xTi_{2-x}(PO_4)_3$ and $Na_3Zr_2Si_2PO_{12}$[13,14,16,36–38]. In Li-NASICON, the transition metal ions are mainly Ge$^{4+}$ and Ti$^{4+}$, with ionic radius of 53 pm and 60.5 pm. While in Na-NASICON, the transition metal changes to Zr$^{4+}$, which has a bigger radius of 72 pm compared to Ge$^{4+}$ and Ti$^{4+}$. $Li_{10}GeP_2S_{12}$ and $Na_{11}Sn_2PS_{12}$ are both LISICON type ionic conductors with high ionic conductivity ($10^{-3}$ ~ $10^{-2}$ S/cm), here Sn$^{4+}$ (69 pm) in Na-LISICON is also bigger than Ge$^{4+}$ (53 pm) in Li-LISICON[39,40]. Therefore, selecting larger cations, such as La$^{3+}$ (103 pm), Pr$^{3+}$ (99 pm), and Sm$^{3+}$ (96 pm), could potentially increase the Na$^+$ ionic conductivity of sodium halide SSEs.

Recently, a class of $MCl_3$ (M = La-Gd) based halide SSEs has been reported, in which the Li$^+$ ion conductivity can reach $10^{-3}$ S/cm[41,42]. In this structure, $MCl_9$ tricapped trigonal prisms are stacked along the c-axis, enclosing one-dimensional channels. Li$^+$ ions hop between two neighbored octahedral sites with a short distance of 2.08 Å in this one-dimensional diffusion channel. Similarly, the octahedral sites can also be occupied by Na$^+$ ions, as Lissner et al. reported in 1993 ($Na_{3x}M_{2-x}Cl_6$, M = La-Sm)[43]. As shown in Fig. 1a, Na$^+$ ions partly occupy the octahedral sites along the c-axis, with a slightly larger distance of 2.19 Å between neighbored sites, making $Na_{3x}M_{2-x}Cl_6$ (M = La-Sm) a potential Na$^+$-ion superionic conductor.

In this work, we prepared and optimized the Na$^+$ ion diffusion channel in $NaLaCl_4$ ($Na_{0.5}(Na_{0.25}La_{0.75})Cl_3$) with Zr$^{4+}$ doping ($Na_{1-x}La_{1-x}Zr_xCl_4$ or $Na_{0.5-0.75x}(Na_{0.25}La_{0.75-0.75x}Zr_{0.75x})Cl_3$), which achieved a high ionic conductivity of $2.9 \times 10^{-4}$ S cm$^{-1}$ (30 °C). Detailed structure characterizations and theoretical simulations indicate that the doping of Zr$^{4+}$ expands the sodium-ion migration pathway, thereby promoting fast ion conduction along the c axis. In addition, the oxidation stability of $Na_{1-x}La_{1-x}Zr_xCl_4$ is on par with other types of halide SSEs. The ASSB using $Na_{1-x}La_{1-x}Zr_xCl_4$ as catholyte shows excellent electrochemical performance, which can retain about 94 mA h g$^{-1}$ capacity at 1 C current and 88% capacity after 70 cycles at 0.3 C.

## Result and discussion

### Synthesis and characterizations of $Na_{1-x}La_{1-x}Zr_xCl_4$

High-temperature sintered $NaLaCl_4$ (NLC-HT) exhibits the same structure as $LaCl_3$ ($P6_3/m$), and the synthesized material did not contain detectable impurities (Supplementary Fig. 1). This material can be considered as Na$^+$ doped $LaCl_3$ ($Na_{0.5}(La_{0.75}Na_{0.25})Cl_3$), in which 25% of La$^{3+}$ in the 2c site is replaced by Na$^+$, and the rest Na$^+$ takes the interstitial site 2b. Rietveld refinement (Supplementary Fig. 2a and Supplementary Table 1) indicates the lattice parameters of a = b = 7.567 Å and c = 4.346 Å. The enriched Na$^+$ vacancies at 2b sites and the short distance (2.19 Å) between adjacent sites could potentially benefit the rapid migration of Na$^+$. However, the ionic conductivity of NLC-HT is as low as $10^{-9}$ S cm$^{-1}$, and the migration activation energy is as large as 0.65 eV. The main reason for the low ionic conductivity of $NaLaCl_4$ could come from the large migration barrier for Na$^+$. In other words, the migration path for Na$^+$ ion diffusion is too narrow. Thus, we adopted Zr$^{4+}$ doping at the La$^{3+}$ site (2c) to broaden the diffusion channel. In addition, the higher valence state of Zr$^{4+}$ compared to La$^{3+}$ can introduce more Na$^+$ vacancies in the structure that will be beneficial to its ionic conductivity.

As shown in Supplementary Figs. 1 and 2, the maximum Zr$^{4+}$ doping amount in $NaLaCl_4$ is around 0.2 ($Na_{0.8}Zr_{0.2}La_{0.8}Cl_4$) without observing significant impurity phases. The a lattice parameter increases from 7.567 Å to 7.573 and 7.578, and then decreases to 7.545 Å with the increase of Zr$^{4+}$ content, The c lattice parameter exhibits similar trends that increases from 4.346 Å to 4.353 Å and 4.357 Å, and then decreases to 4.359 Å (Supplementary Fig. 3, Supplementary Tables 2–4). The lattice parameters increase with the amount of dopant until impurity phases appear at the Zr$^{4+}$ doping level of 0.3, suggesting the successful doping of Zr$^{4+}$ in $NaLaCl_4$. The increase lattice parameters by Zr$^{4+}$ doping decreased the Na$^+$ migration energy to 0.36 eV (NLZC0.2-HT, Supplementary Figs. 4 and 5) that is comparable to other Na$^+$ SSEs[18]. In addition, the ionic conductivity is significantly improved from $10^{-9}$ S cm$^{-1}$ (NLC-HT) to $10^{-6}$ S cm$^{-1}$ (NLZC0.2-HT) although further improvement is still needed for its practical application.

To further promote the ionic conductivity, the high-temperature sintered samples were ball milled to introduce disordering and defects, which can further increase the doping concentration of Zr$^{4+}$ to expand Na$^+$ diffusion channel (NLZCx-HM). As shown in Supplementary Fig. 1b, the impurities in NLZC0.3-HT disappear after ball milling (NLZC0.3-HM) that can be attributed to the improved thermodynamic stability of Zr-doped $NaLaCl_4$ with enriched defects and higher entropy through ball milling. However, when the doping level of Zr$^{4+}$ is increased to 0.4 (NLZC0.4-HM), the impurities cannot be fully removed even after ball milling treatment. Compared to high-temperature sintered samples (NLZCx-HT), the lattice parameter a in NLZCx-HM changes to 7.565 Å (x = 0), 7.570 Å (x = 0.1), 7.570 Å (x = 0.2) and 7.605 Å (x = 0.3) after ball milling, while lattice parameter c increases to 4.372 Å (x = 0), 4.372 Å (x = 0.1), 4.372 Å (x = 0.2) and 4.400 Å (x = 0.3).

The nonlinear change in lattice parameters implies the occurrence of other defects induced by the ball milling process, such as anti-site defects. X-ray diffraction (XRD) refinement results in Fig. 1b–e and Supplementary Tables 5–8 reveal that part of La$^{3+}$ would take the 2b site and further expands the lattice. Synchrotron-based XRD of $NaLaCl_4$-HT and $NaLaCl_4$-HM were carried out to uncover the changes in fine structure and the reason for volume expansion. The refinement results shown in Supplementary Fig. 6 and Supplementary Table 9 reveal a small amount of Na$^+$ at the 2b site (0.16) in $NaLaCl_4$-HT, which means this site is metastable for Na$^+$ occupying. Excess Na exists in the form of unknown impurities (Supplementary Fig. 6). During the ball milling process, the high energy leads to more Na$^+$ occupancy at the 2b site (0.251), resulting in the disappearance of impurities and lattice expansion (Supplementary Fig. 7 and Supplementary Table 10). Moreover, the expansion of the lattice stabilizes the Na$^+$ at the 2b site. Zr$^{4+}$ doping at La$^{3+}$ site can also stabilize Na$^+$ at 2b site and result in more Na$^+$ occupancy at the 2b site (0.461) in $Na_{0.7}La_{0.7}Zr_{0.3}Cl_4$-HT (Supplementary Fig. 8 and Supplementary Table 11), leading to an increase of lattice parameter a (7.583 Å). No significant crystal

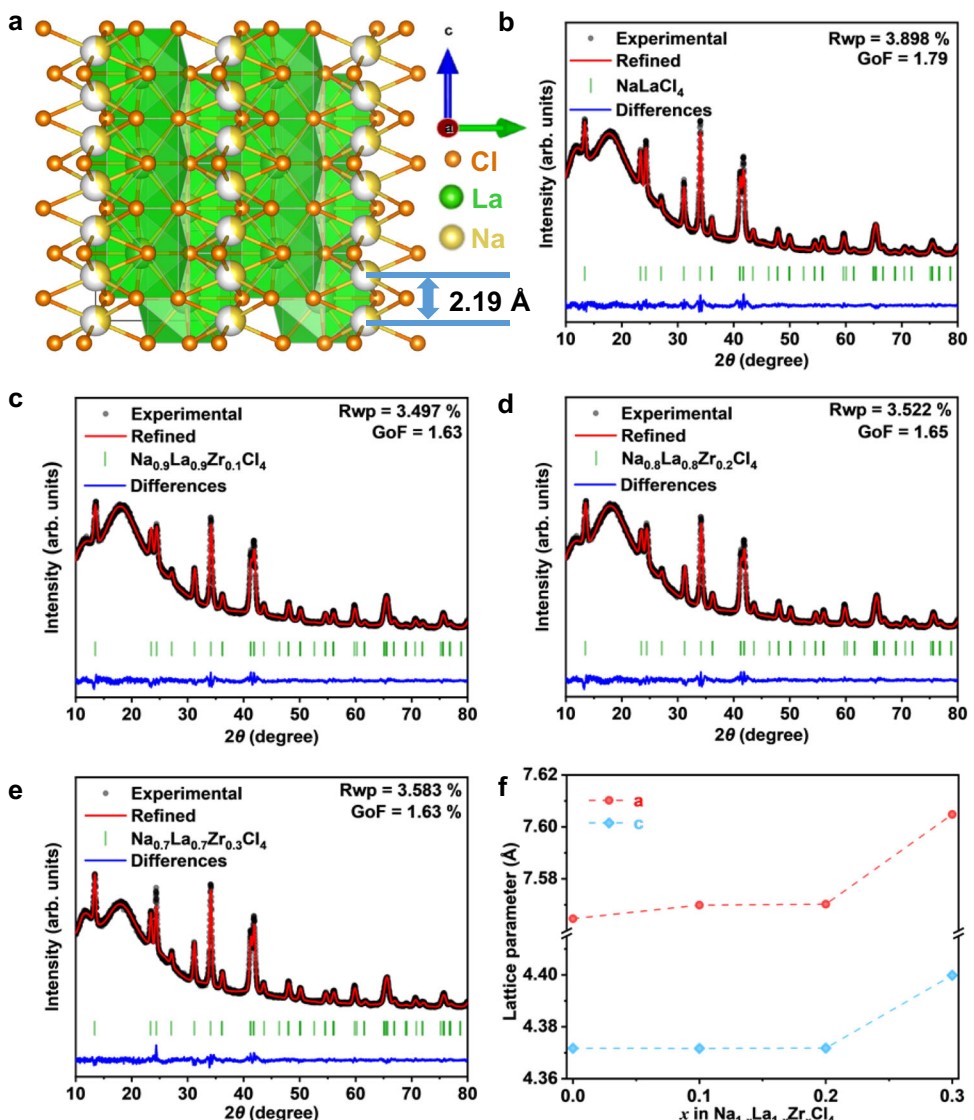

**Fig. 1 | Crystal structure analysis.** Crystal structure of the NaLaCl$_4$ (the white ball: vacancy) (**a**). Rietveld refinements of NaLaCl$_4$-HM (**b**), Na$_{0.9}$La$_{0.9}$Zr$_{0.1}$Cl$_4$-HM (**c**), Na$_{0.8}$La$_{0.8}$Zr$_{0.2}$Cl$_4$-HM (**d**), Na$_{0.7}$La$_{0.7}$Zr$_{0.3}$Cl$_4$-HM (**e**) and lattice parameters of Na$_{1-x}$La$_{1-x}$Zr$_x$Cl$_4$-HM obtained from XRD refinement (**f**).

structural changes were found in the framework, such as the decrease in orderliness (Supplementary Figs. 6 and 7), which could be related to the strong bond energy of M-Cl that helps keep its crystal structure.

### Ion conduction in Na$_{1-x}$La$_{1-x}$Zr$_x$Cl$_4$

The structural change induced by ball milling and the expanded lattice is beneficial for the Na$^+$ ion conduction in Na$_{1-x}$Zr$_x$La$_{1-x}$Cl$_4$. As a result, the ionic conductivity is further increased by one to two orders of magnitude (Fig. 2a, b and Supplementary Fig. 9). Especially for the NLZC0.3-HM sample, its ionic conductivity reaches $2.9 \times 10^{-4}$ S cm$^{-1}$ that is much higher than other sodium halide SSEs reported in literature[4,18,22,23,44]. Moreover, its activation energy reduces to 0.33 eV, lower than the same type SSEs (Fig. 2c). In addition, the electronic conductivity of NLZC0.3-HM is measured to be $1.3 \times 10^{-8}$ S cm$^{-1}$ (Fig. 2d), which is at the similar level compared to other types of SSEs[28].

The significantly increased ionic conductivity with Zr doping (Na$_{1-x}$Zr$_x$La$_{1-x}$Cl$_4$) was studied through theoretical simulations and X-ray absorption fine structure (XAFS). There are abundant channels available for one-dimensional (1D) ion diffusion in rare earth UCl$_3$ (U = La-Sm) halides (Supplementary Fig. 10a) with the $P6_3/m$ lattice. When this 1D channel is occupied by alkali metal ions, such as Li$^+$ (Li$_{3x}$La$_{2-x}$Cl$_6$), Na$^+$

(Na$_{3x}$La$_{2-x}$Cl$_6$ in Supplementary Fig. 10b), they may quickly migrate along this one-dimensional channel. Here, rapid Na$^+$ ions diffusion can be achieved in NLZC by doping Zr$^{4+}$ into NaLaCl$_4$ (Supplementary Fig. 10c). By randomly generating $2 \times 2 \times 3$ supercell structures and conducting structural relaxation (Fig. 3a), the most likely NLZC structures (Fig. 3b) were selected based on the criteria of energy and unit cell distortion. With the doping of Zr$^{4+}$ at the La$^{3+}$ site, the higher valence and smaller size of Zr$^{4+}$ (0.89 Å vs. 1.216 Å for La$^{3+}$) leads to shorter M-Cl bond (2.64 Å for Zr and 2.96 Å for La). As a result, the bond length of NaCl in the 1D channel increases from the original 2.87 Å to 2.91 Å (Fig. 3c). The longer NaCl bond broadens the diffusion bottleneck in the 1D channel, lowers the site energy of Na$^+$ ions at the bottleneck, thereby lowering their migration activation energy and increasing the Na$^+$ ion conductivity. The AIMD simulations imply that the migration of Na$^+$ ions in NLZC requiring a lower barrier energy comparing to NLC (0.04 eV in NLZC vs. 0.26 eV in NLC, Fig. 3d). It has been found that the ion diffusion coefficient along the $c$ direction is two orders of magnitude higher than along the $ab$ plane for much lower migration barrier in previous studies[41], suggesting that NLZC is a 1D conductor.

For this 1D conductor, ion mixing along the diffusion channel should have a significant impact on the Na$^+$ ion conduction. A small

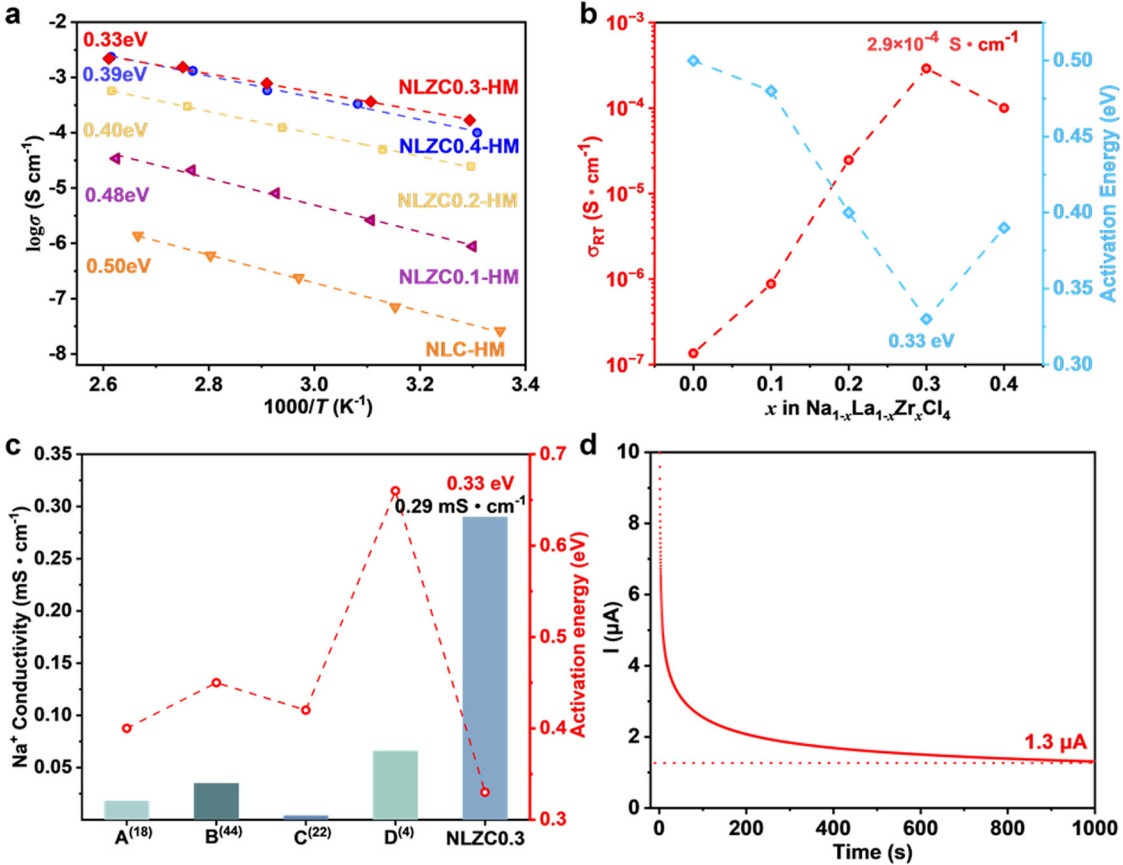

**Fig. 2 | Electrochemical properties of NLZCx-HM. a** Arrhenius conductivity plots of NLZCx-HM ($0 \leq x \leq 0.4$), **b** ionic conductivities and activation energies of NLZCx-HM ($0 \leq x \leq 0.4$), **c** ionic conductivity and activation energy comparing with

$Na_2ZrCl_6$ (A)[18], $Na_{2.4}Er_{0.4}Zr_{0.6}Cl_6$ (B)[44], $NaAlCl_4$ (C)[22], $Na_{2.25}Y_{0.25}Zr_{0.75}Cl_6$ (D)[4] from literature, **d** DC polarization curve of $Na_{0.7}La_{0.7}Zr_{0.3}Cl_4$-HM under 0.5 V.

amount of $La^{3+}$ (0.06) occupy the *2b* site along the c axis, due to the similar radius of $La^{3+}$ (1.216 Å) and $Na^+$ (1.24 Å with 9 coordinations) (Supplementary Table 8). The high valence of $La^{3+}$ enhances the La-Cl bond energy and decreases its mobility, thus the $La^{3+}$ on the migration channel should block the migration of $Na^+$. Similar phenomena have also been found in other one-dimensional conductors, such as $LiFePO_4$[45]. Thus, the effect of $Na^+$/$La^{3+}$ mixing on sodium-ion conduction is also simulated. To save the computation time, a larger mixing ratio of 0.17 was constructed in $2 \times 2 \times 3$ supercell. AIMD was used to study the change of 1D ionic conductivity before and after doping $Zr^{4+}$. The NLZC-La/Na-mixing structure was constructed with energy as the criterion (Supplementary Fig. 11). The thermodynamic stability of the constructed model was also investigated (Supplementary Fig. 12). The comparison of results shows that when the NLZC is at 400 K, the structural frame with $LaCl_3$ as the premise has greater distortion, which means it is more unstable, while the NLZC-La/Na-mixing still keeps the structure relatively stable under 500 K, indicating that the mixing makes the material more stable. The ionic conductivity of NLZC is $13.84 \times 10^{-3}$ S cm⁻¹ at 300 K, while that of NLC is only $1.15 \times 10^{-9}$ S cm⁻¹, which explains the improvement of ionic conductivity caused by $Zr^{4+}$ doping as shown in Fig. 3d. Considering the La/Na mixing at *2b* site (NLZC-La/Na mixing), the ionic conductivity decreases to $5.93 \times 10^{-3}$ S cm⁻¹ at 300 K. In addition, this atomic-mixing leads to a larger migration barrier (0.21 eV) comparing to that without atomic-mixing (0.04 eV). Moreover, the Na-ion migration of NLZC is much higher than that of NLC and NLZC-La/Na mixing through the probability density distribution (Fig. 3e and Supplementary Fig. 13) of AIMD.

The Zr *K*-edge and La *L₃*-edge XAFS were obtained to reveal the local coordination structure of $Na_{0.7}La_{0.7}Zr_{0.3}Cl_4$-HT, $Na_{0.7}La_{0.7}Zr_{0.3}Cl_4$-HM and $NaLaCl_4$-HT. It can be seen that the absorption edge of the Zr *K*-edge and La *L₃*-edge of three basically coincide (Supplementary Fig. 14), indicating equal valence with $Zr^{4+}$ and $La^{3+}$. From the R-space of Zr *K*-edge, the main peak at about 2 Å could be recognized as Zr-Cl coordination from $Na_{0.7}La_{0.7}Zr_{0.3}Cl_4$-HT and $Na_{0.7}La_{0.7}Zr_{0.3}Cl_4$-HM (Supplementary Fig. 15). Similarly, the R-space curve of La *L₃*-edge also show a significant difference in the position of the main peaks compare with $La_2O_3$ (Supplementary Fig. 16), which could be recognized as La-Cl coordination of $NaLaCl_4$, $Na_{0.7}La_{0.7}Zr_{0.3}Cl_4$-HT and $Na_{0.7}La_{0.7}Zr_{0.3}Cl_4$-HM. As a result, Zr-Cl bond and La-Cl bond lengths are different in the same sample.

As shown in Fig. 4, EXAFS fitting was performed to quantitatively compare the Zr-Cl and La-Cl coordination structures in $Na_{0.7}La_{0.7}Zr_{0.3}Cl_4$, and the results are summarized in Supplementary Table 12. $NaLaCl_4$, NLZC0.3-HT, and NLZC0.3-HM have a similar bond length of La-Cl and the coordination numbers of La, which proves that the replacing of the Zr element and ball milling process does not change its structure, which is consistent with the XRD results (Supplementary Fig. 1). It can be confirmed that the bond length of Zr-Cl is about 2.48 Å while it is 2.94 Å for La-Cl. These results confirm that the doping of Zr in the La site shortens the bond length of M-Cl, which broadens the diffusion path of $Na^+$ ions along the c axis, as revealed from theoretical simulations (Fig. 3). In addition, the coordination numbers of Zr and La are close, further confirming the successes doping of Zr at the La site, although the radius of $Zr^{4+}$ is significantly different from $La^{3+}$.

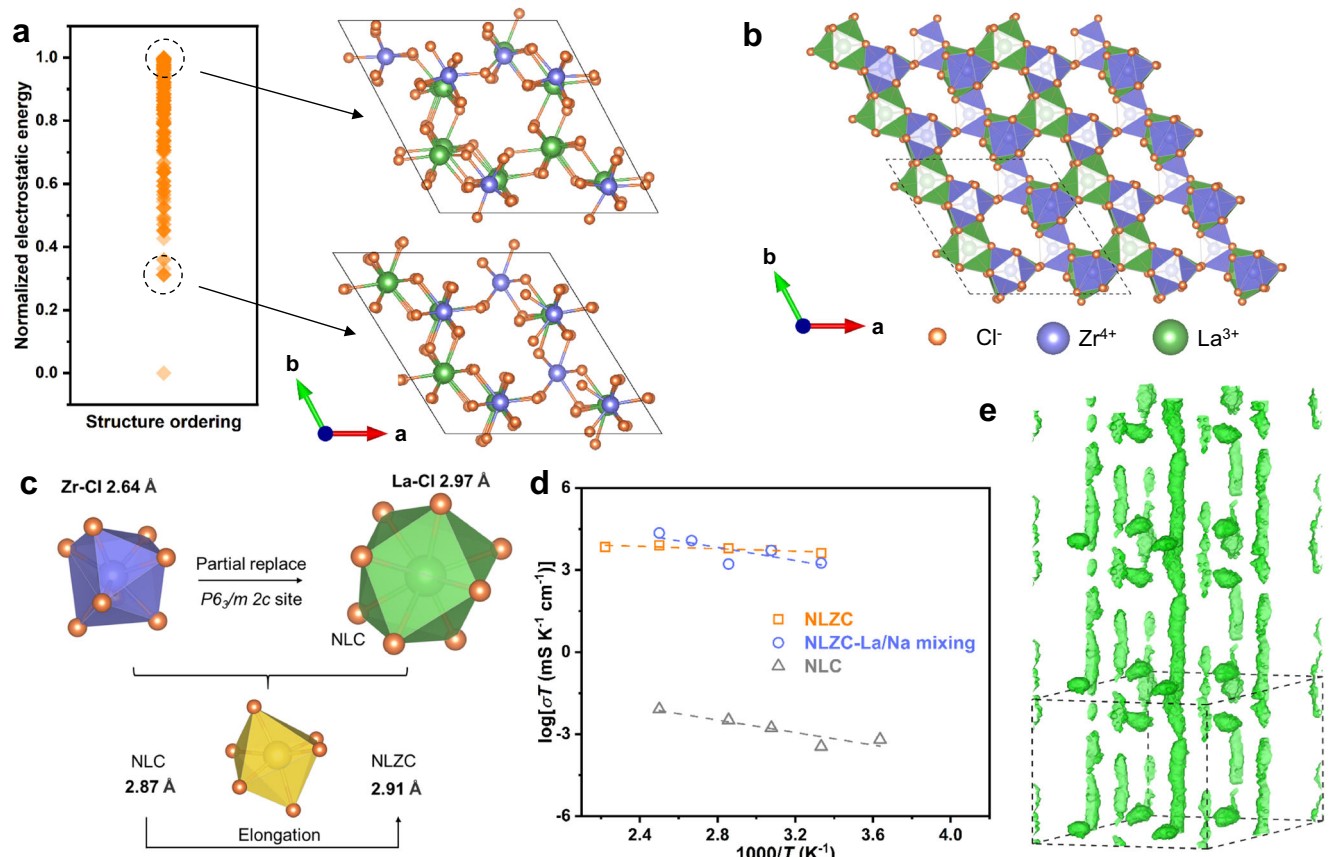

**Fig. 3 | Na$^+$ migration mechanism of NLZCx-HM. a** Normalized electrostatic energy of 500 Na-free configurations in NLZC supercell model with representative high-energy (upper) and low-energy (lower), and corresponding results of the relaxation of two of the structures (Na ions were removed for clearer exhibition). **b** 221 supercell of NLZC for better displaying the *c* axis channel. **c** bond length for La-Cl, Zr-Cl, and NaCl in NLC or NLZC. **d** Arrhenius plot of Na$^+$ migration pathways. Diffusivity in the NLZC lattice from AIMD simulations. **e** Na$^+$ probability density, represented by green iso-surfaces from AIMD simulations at 300 K in the NLZC lattice.

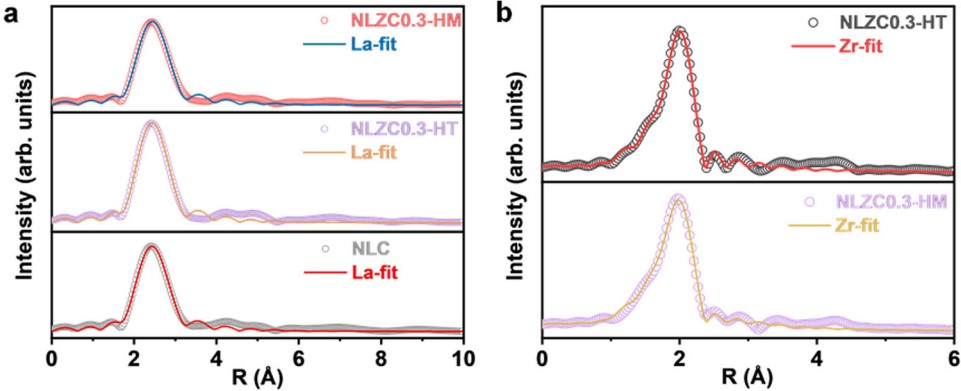

**Fig. 4 | R-space fitting curves of Na$_{0.7}$La$_{0.7}$Zr$_{0.3}$Cl$_4$. a** La-Cl coordination structures. **b** Zr-Cl coordination structures.

## Electrochemical performance of ASSBs using NLZC0.3-HM

The typical electrochemical stable window of sodium halide SSE is from 1.5 V to 3.7 V[4]. Thus, sodium halide SSEs are mainly used as catholytes to match high-potential cathodes. Linear scanning voltammetry (LSV) of NLZC0.3-HM-Super P||NLZC0.3-HM||Na$_3$PS$_4$||Na$_2$Sn reveals that the electrochemical stable window of NLZC0.3-HM is from 1.33 V to 3.80 V *vs.* Na$_2$Sn at 30 °C and from 1.44 V to 3.79 V *vs.* Na$_2$Sn at 60 °C (Supplementary Fig. 17), which is consistent with the theoretical results of sodium halide SSEs[4]. The high reduction potential of NLZC0.3-HM indicates that this SSE is thermodynamically unstable

against Na$_2$Sn or Na metal anodes. To verify the dynamic stability of NLZC0.3-HM against low potential anodes and its possibility as an anolyte, symmetric cell of Na$_2$Sn||NLZC0.3-HM||Na$_2$Sn was assembled and cycled (Supplementary Fig. 18). When cycled at current density with 0.01 mA cm$^{-2}$, the voltage increases to 0.1 V within only 70 h, showing the instability of Na$_2$Sn/NLZC0.3-HM interface (Supplementary Fig. 18). Both La *3d* and Zr *3d* peaks shift to the low-energy direction, indicating the reduction of Zr$^{4+}$ and La$^{3+}$ after cycling (Supplementary Fig. 19). Thus, this SSE cannot be directly used as an anolyte.

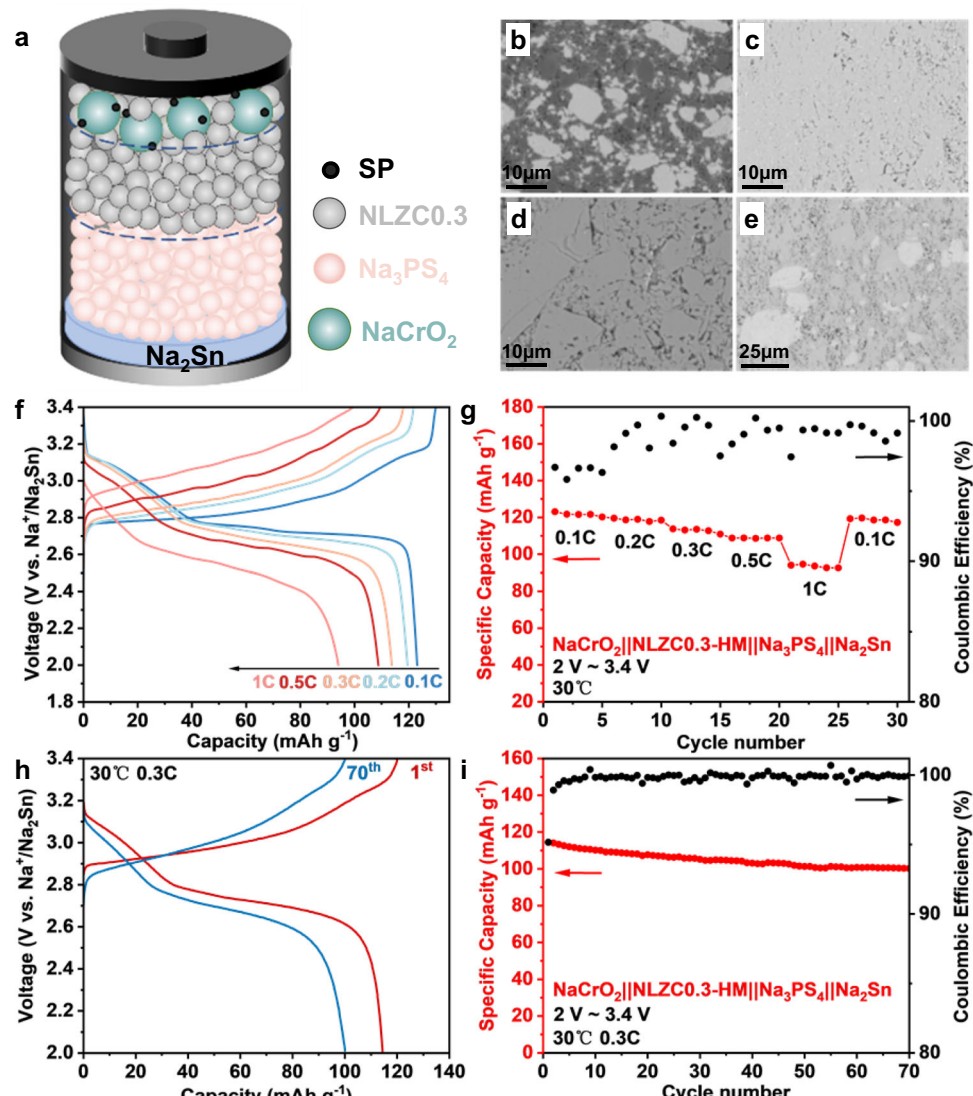

**Fig. 5 | All-solid-state batteries performance of the NaCrO₂∥NLZC0.3-HM∥Na₃PS₄∥Na₂Sn. a** Configuration of ASSB with NLZC0.3-HM as SSEs. Cross-section SEM for NaCrO₂ composite (**b**), Na₀.₇La₀.₇Zr₀.₃Cl₄ SSE (**c**), Na₃PS₄ SSE (**d**), and Na₂Sn anode (**e**) in the ASSB. **f** charge-discharge profiles at different currents (0.1 C, 0.2 C, 0.3 C, 0.5 C, 1 C between 2 and 3.4 V vs. Na/Na⁺, 30 °C), **g** rate performance, **h** charge–discharge profiles at different cycles (0.3 C, 30 °C), **i** cycle performance.

ASSBs were assembled with homemade NaCrO₂ (Supplementary Fig. 20) composite cathode, Na₂Sn anode, and NLZC0.3-HM SSE, with Na₃PS₄ serving as a transition layer to isolate the NLZC0.3-HM electrolyte from the Na₂Sn anode (Fig. 5a). From the SEM image of the ASSB cross-section (Supplementary Fig. 21), the thickness of these four layers is determined to be ~60 μm (NaCrO₂ composite cathode), ~300 μm (NLZC0.3-HM SSE), ~400 μm (Na₃PS₄ SSE) and ~70 μm (Na₂Sn anode) respectively. Enlarged SEM image (Fig. 5b) and EDS (Supplementary Fig. 22) confirm the tight physical contact between NaCrO₂ cathode and NLZC0.3-HM SSE, which should be beneficial to the Na⁺ ion diffusion at the interface. The two SSEs layers and Na₂Sn anode layer are quite dense with only small amount pores (Fig. 5c–e). The boundaries between layers are relatively clear, with little mixing between them (Supplementary Figs. 22–25). The total resistance of these ASSBs before cycling is about 400 Ω (Supplementary Fig. 26), which is contributed by NLZC0.3-HM (~150 Ω, Supplementary Fig. 6d), Na₃PS₄ SSE (~80 Ω, Supplementary Fig. 27), Na₂Sn anode side (~20 Ω, Supplementary Figs. 27 and 28a), resistance between NLZC0.3-HM and Na₃PS₄ (~50 Ω, Supplementary Fig. 28a, b), and interface from NaCrO₂ cathode side (~100 Ω, Supplementary Figs. 26 and 28b). Na₂.₉PS₃.₉Cl₀.₁ with higher ionic conductivity (Supplementary Fig. 29)

was also selected as a transition layer to reduce the total resistance of ASSB. However, this electrolyte reacts with NLZC0.3-HM, and the resistance keeps increasing when co-pressed together (Supplementary Fig. 30). The XPS spectra indicate structural changes with NLZC0.3-HM and NPSC mixing together (Supplementary Fig. 31), especially for NPSC, which causes an increase in resistance (Supplementary Fig. 30).

The ASSB of NaCrO₂∥NLZC0.3-HM∥Na₃PS₄∥Na₂Sn was cycled between 2.0 V and 3.4 V, in which range the NLZC0.3-HM SSE is electrochemically stable. When cycling at 0.1 C (30 °C), the discharge capacity for the first cycle reaches 123 mA h g⁻¹, with a high Columbic efficiency (CE) of 95%, reflecting a highly irreversible reaction in this ASSB. When current density increases, the capacity decreases from 123 mA h g⁻¹ (0.1 C) to 119 mA h g⁻¹ (0.2 C), 114 mA h g⁻¹ (0.3 C), 108 mA h g⁻¹ (0.5 C), and 94 mA h g⁻¹ (1 C), showing high-capacity retention at high current density (Fig. 5f, g). Compared to other halide SSEs with lower ionic conductivity, the rate performance of ASSB using NLZC0.3-HM is better[4,46]. After cycling at 0.3 C for 70 times, the capacity retains 100 mA h g⁻¹, which is 88% of the initial capacity (Fig. 5h, i). Part of the capacity loss should come from an increase in resistance, which is 780 Ω after 70 cycles at 0.3 C (Supplementary Fig. 26). This increase in resistance may originate from Na₃PS₄/Na₂Sn

interface, and further enhancement in cycle performance can be achieved with other SSE transition layers, such as $Na_4(B_{12}H_{12})(B_{10}H_{10})$[46]. However, this is still much better than ASSB using sulfide catholyte (10% capacity retention after only 20 cycles at 0.1 C), indicating much better stability of NLZC0.3-HM (Supplementary Fig. 32).

In summary, a sodium halide SSE of $Na_{0.7}La_{0.7}Zr_{0.3}Cl_4$ with a space group of $P6_3/m$ was designed and synthesized. In this structure, $La^{3+}$, $Zr^{4+}$ and partly $Na^+$ occupied a tricapped trigonal prisms site ($MCl_9$) and stack along the $c$-axis, in which the one-dimensional channel is enclosed between them. The rest of $Na^+$ ions take the sites in this one-dimensional channel and diffuse along the channels, giving high ionic conductivity. Structure characterizations and theoretical simulation results reveal that the doping of $Zr^{4+}$ in the $La^{3+}$ site shortens the length of the M-Cl bond ($M = Zr^{4+}$, $La^{3+}$), which broaden $Na^+$ ion diffusion path and improves $Na^+$ ion conductivity. On the other hand, a small amount of $La^{3+}$ in the one-dimensional diffusion path blocks $Na^+$ ion diffusion, which reduces the ionic conductivity. As a result, $Na_{0.7}La_{0.7}Zr_{0.3}Cl_4$ achieves a decent ionic conductivity of $2.9 \times 10^{-4}$ S cm$^{-1}$ (30 °C) and a low activation energy of 0.33 eV. Finally, the electrochemical stable window of $Na_{0.7}La_{0.7}Zr_{0.3}Cl_4$ is determined to be as high as 3.7 V $vs.$ Na/Na$^+$, enabling high stability against the $NaCrO_2$ cathode. Promising electrochemical performance of $NaCrO_2/Na_2Sn$ all-solid-state cells using $Na_{0.7}La_{0.7}Zr_{0.3}Cl_4$ as the catholyte is achieved, with high initial CE of 95% and 88% capacity retention after cycling at 0.3 C for 70 times. In addition, a high capacity of 94 mA h g$^{-1}$ can be obtained at 1 C current.

## Methods

### Material synthesis

The raw materials of NaCl, $ZrCl_4$, and $LaCl_3$ were all purchased from Sinopharm without further purification. NaCl, $ZrCl_4$, and $LaCl_3$ were hand milled with a Na: Zr: La: Cl molar ratio of (1-$x$): $x$: (1-$x$): 4 for 30 min. The pre-mixed powders were then sintered at 450 °C for 10 h under an Ar environment. After that, the samples were ball-milled (QM-3SP2, Nanda Instruments) in a 50 ml $ZrO_2$ jar, with a speed of 550 rpm for 10 h under vacuum. All operations were performed in an Ar-filled glovebox (Mikrouna, $H_2O < 0.1$ ppm, $O_2 < 0.1$ ppm).

$Na_{2.9}PS_{3.9}Cl_{0.1}$ powders were prepared according to our previous reports[47]. A stoichiometric mixture of $Na_2S$ (Sigma Aldrich), $P_2S_5$ (99%, Sigma Aldrich), and NaCl (Sinopharm) was ball milled at 550 rpm for 2 h and sintered at 300 °C for 2 h under Ar. After ground again, the powders were sintered at 420 °C for another 12 h under Ar. $Na_3PS_4$ electrolytes were prepared via mechanochemical method. $Na_2S$ (Sigma Aldrich) and $P_2S_5$ (Sigma Aldrich) with a stoichiometric ratio were ball-milled at 500 rpm for 12 h. The mixed powders were vacuum sealed in a quartz tube and annealed at 280 °C for 1 h.

The $NaCrO_2$ cathode was prepared via a solid-state reaction. $Na_2CO_3$ and $Cr_2O_3$ (Sinopharm) were mixed with a Na:Cr ratio of 1.05:1, 5% excess Na to compensate for the volatilization at high temperatures. The mixture was first ball milled at 550 rpm for 10 h under vacuum and then sintered at 900 °C for 10 h (OTF-1200X, HF-Kejing) under Ar. When cooling to 200 °C, the sample was transferred to the glovebox quickly to reduce sample exposure to air.

$Na_2Sn$ anode was prepared with ball milling Na and Sn metal. The stoichiometric mixture of Na and Sn metal (Sinopharm) was first rolled together and then transferred into a steel jar, sealed under Ar. The mixture was ball milled at 300 rpm for 10 h to obtain the homogenous product. If the product is not homogenous, another 10 h ball milling is needed.

### Characterizations

Powder XRD experiments were performed using a Philips X'Pert powder diffractometer (D/MAX2500VL/PC, Rigaku) at 45 kV and 40 mA with Cu-K$_\alpha$ radiation ($\lambda = 1.5406$ Å). The samples were placed in a zero-background holder and sealed with a Kapton film to avoid air exposure. The data was collected at room temperature with 2θ from 10° to 80°.

Rietveld refinement of $Na_{1-x}Zr_xLa_{1-x}Cl_4$ were carried out from XRD data with strong intensity. The following parameters were refined step wisely: (1) scale factor, (2) background using linear interpolate function with 10 coefficients, (3) peak shape using the pseudo-Voigt function, (4) unit cell parameters and fractional atomic coordinates, (5) fractional occupancy and thermal displacement parameters (Uiso).

Synchrotron XRD experiments were carried out at XPD beamline (X-ray Powder Diffraction, ID28) at the National Synchrotron Light Source II (NSLS-II), Brookhaven National Laboratory, USA, with a photon wavelength of 0.185794 Å.

X-ray photoelectron spectroscopy was performed using a PHI 5000 VersaProbe III with a monochromatic Al Kα X-ray source.

The X-ray absorption spectra (XAS) including X-ray absorption near-edge structure (XANES) and extended X-ray absorption fine structure (EXAFS) of the sample at Zr $K$-edge and La $L3$-edge collected at Beijing Synchrotron Radiation Facility 1W1B beamline. Samples were covered with Kapton film to avoid air exposure.

The ionic conductivity of $Na_{1-x}Zr_xLa_{1-x}Cl_4$ was measured at 25 °C using Ac electrochemical impedance spectroscopy (EIS) in the frequency range from 1 MHz to 1 Hz with a potential perturbation of 50 mV (Biologic SP-200). The activation energy was calculated based on variable-temperature impedance from room temperature up to 100 °C in a microclimate chamber. The as-synthesized $Na_{1-x}Zr_xLa_{1-x}Cl_4$ powders were pressed into pellets with a diameter of 12 mm, under a pressure of 260 MPa, and then sandwiched by two steel rods for all measurements.

The electrochemical stability of $Na_{0.7}La_{0.7}Zr_{0.3}Cl_4$ was characterized with cyclic voltammetry. $Na_{0.7}La_{0.7}Zr_{0.3}Cl_4$ was mixed with 30% SP and used as the cathode, $Na_3PS_4$ was used as the SSE, and $Na_2Sn$ was used as the anode. These three parts were respectively pressed under 300 MPa in a cylinder cell with a diameter of 12 mm. The cyclic voltammetry was carried on Biologic SP-200 with a scan rate of 0.2 mV/s and voltage window of 1.3–4.9 V.

All-solid-state $NaCrO_2$||NLZC0.3-HM||$Na_3PS_4$||$Na_2Sn$ battery was fabricated using the following procedure. The composite cathode was made by ball milling the mixture of $NaCrO_2$, NLZC0.3-HM, and SP (50: 50: 3 in weight ratio) at 300 rpm for 30 min. The ASSB was made by co-pressing $Na_2Sn$ anode (50 mg), $Na_3PS_4$ (70 mg) anolyte, NLZC0.3-HM (70 mg) catholyte, and $NaCrO_2$ composite cathode (12 mg) together in order, and under 300 MPa. The ASSB was cycled between 2.0 and 3.4 V vs. Na/Na$^+$, under current densities of 0.1 C, 0.2 C, 0.3 C, 0.5 C, 1 C, respectively. The specific capacity is calculated based on the mass of $NaCrO_2$ cathode.

SEM image of the assembled ASSBs were conducted on a higher solution field emission scanning electron microscope (Hitachi). Before that, the ASSBs were polished using a cooling cross-section polisher (JEOL) to obtain a smooth surface.

### Density functional theory calculations

The first-principles calculations were conducted using the Vienna Ab initio Simulation 5.4.4 Package based on density functional theory[48,49]. The exchange-correlation interaction was generalized gradient approximation (GGA) using the Perdew-Burke-Ernzerhof (PBE) exchange-correlation functional while the projector augmented-wave (PAW) pseudopotential was used to account electron-ion interactions[50–52]. The cutoff energy for the plane-wave basis was set to 400 eV, and the total energy convergence was set to be lower than $1 \times 10^{-5}$ eV, with the force convergence at 0.03 eV Å$^{-1}$, the integrations in the reduced Brillouin zone are performed on a Γ-centered $3 \times 3 \times 3$ special k-points for all calculations.

## Ab initio molecular dynamics simulations

In order to realize the fractional occupied structures, the structure of 2 × 2 × 3 supercell of NLZC0.3 and NLC was randomly generated using supercell software package, and the lowest electrostatic potential 500 structures were output, structural convergence was carried out respectively. The optimal structure was selected based on lattice difference and energy. Considering that La/Na mixing appears in the *2c* site of the synthesized materials, the mixing structure is selected based on the criteria of energy and lattice difference, mixing structure were also screened in 2 × 2 × 3 supercell for reducing the time consuming.

The AIMD simulation of the supercell model was carried out to study ion diffusion, and the non-spin polarization model was used. The SCF convergence criterion of $10^{-4}$ eV in the NVT ensemble lasted above 40 ps with a time step of 2 fs until the diffusivity (D/cm$^2$ s$^{-1}$) converges[53]. *D* is calculated as the mean azimuth shift (*MSD*) over the time interval *Δt*:

$$D = \frac{1}{2Nd\Delta t} \sum_{i=1}^{n} \langle |r_{Na}(t+\Delta t) - r_{Na}(t)|^2 \rangle \tag{1}$$

where *N* is the total number of diffusion ions, *d* is the dimension of the diffusion system, $r_{Na}(t)$ is the displacement of the *Na*-th ion at time *t* and the bracket represents averaging over *t*. Ionic conductivity (*σ/S cm$^{-1}$*) was derived from the Nernst–Einstein relationship:

$$\sigma = \frac{nz^2e^2}{K_BT} D \tag{2}$$

Where *n* is the number of diffusion ions per unit volume (Å$^3$), *z* is ionic charge number, *e* is the electron charge (*C*).

## Data availability

The data that support the findings of this study are available within the article (and its Supplementary Information files) and from the corresponding authors upon reasonable request. Source data are provided with this paper.

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

## Acknowledgements

This study was supported by the National Natural Science Foundation of China (U2330101 to X.F., 52072105 to H.X., 12305368 to W.X. and 22073087 to X.W.), the National Natural Science Foundation for Distinguished Young Scholars (22225301 to X.W.), the Anhui Provincial Natural Science Foundation (2108085J23 to H.X.), the Major Science and Technology Projects in Anhui Province (202203a05020032 to X.F., 2022e03020004 to X.F., 202003a05020014 to H.X. and 2021e03020001 to H.X.), the CAS Project for Young Scientists in Basic Research (YSBR-004 to X.W.), the Strategic Priority Research Program of the Chinese Academy of Sciences (XDB0450101 to X.W.), the Fundamental Research Funds for the Central Universities (JZ2022HGTB0251 to X.F. and 20720220009 to X.W.). The authors would also thank the Beijing Synchrotron Radiation Facility 1W1B beamline for X-ray Absorption Fine Structure characterization and the Super Computer Center of USTCSCC and SCCAS. This research also used resources of the National Synchrotron Light Source II, a U.S. Department of Energy (DOE) Office of Science User Facility operated for the DOE Office of Science by Brookhaven National Laboratory under Contract No. DE-SC0012704 (A.A.). We acknowledge Dr. Nan Wang at Brookhaven National Laboratory for the help in handling samples for the synchrotron measurement.

## Author contributions

X.F. and H.X. conceived the research. C.F. and W.G. performed the materials synthesis, conductivity measurement, structural characterization, Rietveld refinement, and electrochemical tests. Y.L. and X.W. conducted theoretical simulations. W.X. completed the XAFS analysis. A.A. conducted the synchrotron XRD measurements, and L.S. and J.L. completed the analysis. X.F. wrote the manuscript under the assistance of C.F., L.Z., W.D., and W.W. X.F. and H.X. directed the entire study.

## Competing interests

The authors declare no competing interests.
