## [Peer Review File · Nature Communications]

LaCl₃-Based Sodium Halide Solid Electrolytes with High Ionic Conductivity for All-Solid-State BatteriesREVIEWER COMMENTS

Reviewer #1 (Remarks to the Author):

This review paper summarizes a new halide-based sodium ion conductor $\text{Na}_{1-x}\text{Zr}_x\text{La}_{1-x}\text{Cl}_4$, that has a conductivity in the 10^{-4} S/cm range.

While the methodology in the paper is quite thorough, there are a few concerns that need to be addressed before it can be recommended for publication.

- It is unclear why $\text{Na}_2.9\text{PS}_3.9\text{Cl}_0.1$ was chosen as a barrier layer or anolyte in the all-solid-state battery (ASSB) configuration. Although introducing the halide into the configuration improved the cycling stability compared to using $\text{Na}_2.9\text{PS}_3.9\text{Cl}_0.1$ alone, the capacity retention of 75% after 70 cycles is still severe degradation. A more stable or suitable ASSB cell configuration should be used to demonstrate stability.
- It is unclear why cycling was conducted at 60°C and not at ambient temperature or other temperatures.
- While a higher conducting sodium halide improves its usability, its electrochemical window is similar to other reported halides (chlorides). I do not think that supports the claim that higher voltage solid-state batteries can result.
- I do not think the EXAFS data is reliable if the halide has hydrolyzed, as that degradation will fundamentally alter its properties. Air-sensitive sample transfer is a must.

In addition, there are concerns about the novelty of this manuscript.

- A previous study has already elucidated the effect of Zr^{4+} aliovalent doping into sodium halide solid-state electrolytes. <https://www.nature.com/articles/s41467-021-21488-7>. The conclusions in this paper are very similar.
- There is a preprint from 2022 which has shown higher Na conductivity and longer cycling stability than the findings in this manuscript. <https://chemrxiv.org/engage/chemrxiv/article-details/637a89cf20798134fe2e6586>

Reviewer #2 (Remarks to the Author):

In the manuscript "New Sodium Halide Solid Electrolytes with High Ionic Conductivity for All-Solid-State Batteries", the authors reported a new LaCl_3 -based solid electrolyte with ionic

conductivity of 2.9×10^{-4} S/cm. When combined with Na₂Sn anode and NaCrO₂ cathode, the solid-state cell gives an initial capacity of 119 mA h/g at 0.1C. There are quite a few aspects that should be clarified in the manuscript. Some suggestions to further improve the paper:

1 The authors mentioned that “As a result, the bond length of Na-Cl in the 1D channel increases from the original 2.87 Å to 2.91 Å (Fig. 3c), implying that the migration of Na⁺ ions requiring a lower barrier comparing to NLC (0.04 eV in NLC vs. 0.26 eV in NLZC, Fig. 3d).”, which may be not accurate. More explanations about the difference between NLZC and NLZC-La/Na mixing should be given. In actual situations, whether La/Na mixing occurs? If occurs, giving the simulated result about NLZC is reasonable or not?

2 In Table S6 2b site, the occupancy of Na and La is 0.391 and 0.071. But the authors mentioned that “A mixing ratio (0.06) of La/Na at 2b site along the c axis was found from XRD refinement, due to the similar radius of La³⁺ and Na⁺ (Table S6).” How can the authors give the numbers?

3 In absorption edge of the Zr K-edge, the comparison of ZrCl₄ samples should be considered. The comparison of ZrO₂ samples to prove the presence of Zr-Cl coordination is not convincing enough. The XAFS test is capable of avoiding hydrolysis. The fitting results of Zr-O and La-O should be excluded.

4 In Supplementary Figure 12, how to get the oxidation potential of 3.7 V? It seems a clear deviation.

5 In cell tests, why test under 60 °C? Is it reasonable to test at high temperature when SE conductivity is high enough. Typically, lithium cells could test under room temperature with such ionic conductivity of SE.

6 The stability of the new SE against NaSn is not good. However, the LaCl₃-based Lithium SE is stable with Li metal anode. The authors need give more insightful investigation.

Reviewer #3 (Remarks to the Author):

In this manuscript, the authors report on Na_{1-x}Zr_xLa_{1-x}Cl₄ prepared by annealing and subsequent mechanochemical milling method as a halide solid electrolyte material with 0.29 mS cm⁻¹ for all-solid-state Na⁺ batteries. The structure of Na_{1-x}Zr_xLa_{1-x}Cl₄ was characterized using XRD Rietveld refinement and EXAFS. It is interesting that a report about

Na halide electrolyte with hexagonal structure is the first time. Besides, the electrochemical performance of $\text{Na}_{1-x}\text{Zr}_x\text{La}_{1-x}\text{Cl}_4$ was also characterized by cyclic voltammetry and half cell tests. Including above issues, the following discussion points should be addressed extensively.

1. Regarding the composition of ' $\text{Na}_{0.7}\text{Zr}_{0.3}\text{La}_{0.7}\text{Cl}_4$ ': Given that ZrCl_4 has melting and boiling points of 437 °C and 331 °C, respectively, the annealing protocol, which is performed at 450 °C for 10 h in an Ar furnace, may pose a possibility of ZrCl_4 precursor sublimation prior to the intended reaction, potentially altering the nominal compositions.
2. The pivotal role of mechanochemical milling, which enhances conductivity by nearly two orders of magnitude, is broadly, limitedly, and vaguely discussed. Specifically, the claim "To further promote the ionic conductivity, the high temperature sintered samples were ball milled to introduce disordering and defects, thus rising the doping concentration of Zr^{4+} and further expanding the lattice and diffusion channel (NLZCx-HM)." Warrants a meticulous discussion. Considering that the reported material has rapid 1D Na ion conduction channels along the c axis, introducing disorder might adversely impact the 1D connectivity, which necessitates an extensive and in-depth exploration supplemented by additional data and/or theories.
3. Why does the lattice parameter expand upon although Zr is introduced in the lattice and Na vacancy is formed? The ionic radius of Zr^{4+} is smaller than that of La^{3+} .
4. The electrochemical evaluation of NZLC should be supplemented, as the current state does not show any advantages of high ionic conductivity in NZLC. A comparison of rate performance with alternatives like $\text{Na}_{2.25}\text{Y}_{0.25}\text{Zr}_{0.75}\text{Cl}_6$ could be helpful.
5. Cyclic voltammetry for oxidative and reductive stability should be measured separately at room temperature and 60 °C.
6. In line 221, " $\text{NLZC}_{0.3}\text{-HN-super P} \parallel \text{NLZC}_{0.3}\text{-HM} \parallel \text{Na}_{2.9}\text{PS}_{3.9}\text{Cl}_{0.1} \parallel \text{Na}$ " should be revised to " $\text{NLZC}_{0.3}\text{-HN-super P} \parallel \text{NLZC}_{0.3}\text{-HM} \parallel \text{Na}_{2.9}\text{PS}_{3.9}\text{Cl}_{0.1} \parallel \text{Na}_2\text{Sn}$ ".
7. The statement, "Since 2018, plenty of lithium-ion halide SSEs with high ionic conductivity up to 10^{-2} S/cm have been investigated, such as Li_3YCl_6 , Li_3InCl_6 , Li_3ScCl_6 , Li_2ZrCl_6 , and LiTaOCl_4 et al.^{29–34}" is lacking original references detailing Li_2ZrCl_6 : Adv. Energy Mater. 2021, 11, 2003190 and Li_3ScCl_6 ($\text{Li}_2\text{Sc}_{2/3}\text{Cl}_4$): Energy Environ. Sci. 2020, 13, 2056. Also, strictly, reference 34, concerning Li_3TiCl_6 , cannot be categorized as a source pertaining to halide electrolytes.

8. Figure S21: Experimental details, like whether the “NLZC+NPSC” cell indicates a simple Li-ion blocking symmetric cell with a bilayer or composite pellet, should be explicitly provided. Also, it is important to carefully examine and discuss if any interfacial resistance exist between different electrolytes, especially at 60 oC, supported by appropriate citations.

Response Letter

Title: New Sodium Halide Solid Electrolytes with High Ionic Conductivity for All-Solid-State Batteries

REVIEWER 1:

Referee comment:

This review paper summarizes a new halide-based sodium ion conductor $\text{Na}_{1-x}\text{Zr}_x\text{La}_{1-x}\text{Cl}_4$, that has a conductivity in the 10^{-4} S/cm range. While the methodology in the paper is quite thorough, there are a few concerns that need to be addressed before it can be recommended for publication.

Author response: We thank the reviewer for the positive comments and carefully revised the relevant content accordingly.

Referee Query 1:

It is unclear why $\text{Na}_{2.9}\text{PS}_{3.9}\text{Cl}_{0.1}$ was chosen as a barrier layer or anolyte in the all-solid-state battery (ASSB) configuration. Although introducing the halide into the configuration improved the cycling stability compared to using $\text{Na}_{2.9}\text{PS}_{3.9}\text{Cl}_{0.1}$ alone, the capacity retention of 75% after 70 cycles is still severe degradation. A more stable or suitable ASSB cell configuration should be used to demonstrate stability.

Author response 1: We are sincerely grateful to the referee for their suggestion. The reason we choose $\text{Na}_{2.9}\text{PS}_{3.9}\text{Cl}_{0.1}$ as a barrier layer instead of Na_3PS_4 is the higher conductivity of 10^{-3} S/cm comparing to Na_3PS_4 (10^{-4} S/cm). However, this electrolyte is incompatible to $\text{Na}_{1-x}\text{Zr}_x\text{La}_{1-x}\text{Cl}_4$ and the interface resistance between $\text{Na}_{2.9}\text{PS}_{3.9}\text{Cl}_{0.1}$ and $\text{Na}_{1-x}\text{Zr}_x\text{La}_{1-x}\text{Cl}_4$ keeps increasing. After mixing together and hold for 5 days, X-ray photoelectron spectroscopy (XPS) shows significant structural changes, especially in NPSC. In the revised manuscript, Na_3PS_4 was chosen as the barrier layer and the stability and cycle life are both enhanced. The changes are detailed in the Response Letter and are highlighted in blue font in the revised main text and figure captions:

Supplementary Figure 27 Nyquist plots of Na_3PS_4 (1 mm in thickness).

Supplementary Figure 30 Nyquist plots of NLZC0.3-HM and $\text{Na}_{2.9}\text{PS}_{3.9}\text{Cl}_{0.1}$ double layer SSE.

Supplementary Figure 31 (a) Zr 3d, (b) S 2p, (c) P 3d and (d) La 3d X-ray photoelectron spectroscopy (XPS) spectra of NLZC0.3, $\text{Na}_{2.9}\text{PS}_{3.9}\text{Cl}_{0.1}$ and NLZC0.3-HM+NPSC.

Fig. 5 (a) Configuration of ASSB with NLZC0.3-HM as SSEs, (b) Cross section SEM for different layers in the ASSB: (b1) NaCrO₂ composite cathode, (b2) Na_{0.7}La_{0.7}Zr_{0.3}Cl₄, (b3) Na₃PS₄, (b4) Na₂Sn anode, (c) charge-discharge profiles at different currents (30 °C), (d) rate performance, (e) charge-discharge profiles at different cycles (30 °C, 0.3C), (f) cycle performance.

“All solid-state batteries were assembled with homemade NaCrO₂ (Supplementary Fig. 20) composite cathode, Na₂Sn anode and NLZC0.3-HM SSE, with Na₃PS₄ serving as a transition layer to isolate the NLZC0.3-HM electrolyte from the Na₂Sn anode (**Fig. 5a**).”

“Enlarged SEM image (**Fig. 5b1**) and EDS (Supplementary Fig. 22) confirm the tight physical contact between NaCrO₂ cathode and NLZC0.3-HM SSE, which should be benefit to the Na⁺ ion diffusion at the interface. The two SSEs layers and Na₂Sn anode layer are quite dense with only

small amount pores (**Fig. 5b2-4**). The boundaries between layers are relatively clear, with little mixing between them (Supplementary Fig. 22-25). The total resistance of these ASSBs before cycling is about 400 Ω (Supplementary Fig. 26), which was contributed by NLZC0.3-HM (~ 150 Ω , Supplementary Fig. 6d), Na₃PS₄ SSE (~ 80 Ω , Supplementary Fig. 27), Na₂Sn anode side (~ 20 Ω , Supplementary Fig. 27 and Supplementary Fig. 28a), resistance between NLZC0.3-HM and Na₃PS₄ (~ 50 Ω , Supplementary Fig. 28a, b), and interface from NaCrO₂ cathode side (~ 100 Ω , Supplementary Fig. 26 and Supplementary Fig. 28b). Na_{2.9}PS_{3.9}Cl_{0.1} with higher ionic conductivity (Supplementary Fig. 29) was also selected as a transition layer to reduce the total resistance of ASSB. However, this electrolyte reacts with NLZC0.3-HM and the resistance keeps increasing when co-pressed together (Supplementary Fig. 30). The XPS spectra indicate structural changes with NLZC0.3-HM and NPSC mixing together (Supplementary Fig. 31), especially for NPSC, which causes an increase in resistance (Supplementary Fig. 30).

The ASSB of NaCrO₂||NLZC0.3-HM||Na₃PS₄||Na₂Sn was cycled between 2.0 V and 3.4 V, in which range the NLZC0.3-HM SSE is electrochemically stable. When cycling at 0.1 C (30 °C), the discharge capacity for the first cycle reaches 123 mA h g⁻¹, with a high columbic efficiency (CE) of 95%, reflecting a highly irreversible reaction in this ASSB. When current density increases, the capacity decreases from 123 mA h g⁻¹ (0.1C) to 119 mA h g⁻¹ (0.2C), 114 mA h g⁻¹ (0.3C), 108 mA h g⁻¹ (0.5C), and 94 mA h g⁻¹ (1C), showing high-capacity retention at high current density (**Fig. 5c, d**). Compared to other halide SSEs with lower ionic conductivity, the rate performance of ASSB using NLZC0.3-HM is better.^{4,46} After cycling at 0.3 C for 70 times, the capacity retains 100 mA h g⁻¹, which is 88% of the initial capacity (**Fig. 5e, f**). Part of the capacity loss should come from an increase in resistance, which is 780 Ω after 70 cycles at 0.3C (Supplementary Fig. 26). This increase in resistance may originate from Na₃PS₄/Na₂Sn interface, further enhancement in cycle performance can be achieved with other SSE transition layers, such as Na₄(B₁₂H₁₂)(B₁₀H₁₀).⁴⁶

Referee Query 2:

It is unclear why cycling was conducted at 60°C and not at ambient temperature or other temperatures.

Author response 2: We thank the reviewer for this insightful question. When choosing $\text{Na}_{2.9}\text{PS}_{3.9}\text{Cl}_{0.1}$ as a barrier layer, the high interfacial resistance between $\text{Na}_{2.9}\text{PS}_{3.9}\text{Cl}_{0.1}$ and $\text{Na}_{1-x}\text{Zr}_x\text{La}_{1-x}\text{Cl}_4$ makes the ASSB unable to operate at ambient temperature. In the revised manuscript, Na_3PS_4 was chosen as the barrier layer and the ASSB is operated at ambient temperature. The changes are detailed in the Response Letter and are highlighted in blue font in the revised main text and figure captions:

“The ASSB of $\text{NaCrO}_2\|\text{NLZC0.3-HM}\|\text{Na}_3\text{PS}_4\|\text{Na}_2\text{Sn}$ was cycled between 2.0 V and 3.4 V, in which range the NLZC0.3-HM SSE is electrochemically stable. When cycling at 0.1 C (30 °C), the discharge capacity for the first cycle reaches 123 mA h g^{-1} , with a high coulombic efficiency (CE) of 95%, reflecting a highly irreversible reaction in this ASSB. When current density increases, the capacity decreases from 123 mA h g^{-1} (0.1C) to 119 mA h g^{-1} (0.2C), 114 mA h g^{-1} (0.3C), 108 mA h g^{-1} (0.5C), and 94 mA h g^{-1} (1C), showing high-capacity retention at high current density (Fig. 5c, d). Compared to other halide SSEs with lower ionic conductivity, the rate performance of ASSB using NLZC0.3-HM is better.^{4,46} After cycling at 0.3 C for 70 times, the capacity retains 100 mA h g^{-1} , which is 88% of the initial capacity (Fig. 5e, f). Part of the capacity loss should come from an increase in resistance, which is 780Ω after 70 cycles at 0.3C (Supplementary Fig. 26). This increase in resistance may originate from $\text{Na}_3\text{PS}_4/\text{Na}_2\text{Sn}$ interface, further enhancement in cycle performance can be achieved with other SSE transition layers, such as $\text{Na}_4(\text{B}_{12}\text{H}_{12})(\text{B}_{10}\text{H}_{10})$.⁴⁶”

Referee Query 3:

While a higher conducting sodium halide improves its usability, its electrochemical window is similar to other reported halides (chlorides). I do not think that supports the claim that higher voltage solid-state batteries can result.

Author response 3: We are sorry about the misunderstanding. High voltage sodium ASSBs we mentioned in the abstract doesn't mean voltage over 4 V vs. Na/Na^+ but voltage around 3 V vs. Na/Na^+ , which is incompatible to sulfide SSEs. When using halide SSEs, the low ionic conductivity results in poor rate performance. In this work, the higher ionic conductivity of NLZC0.3-HM improves the rate performance in ASSBs, compared to the results from literature

(*Nat. Commun.* **12**, 1256 (2021)). To avoid misunderstanding, we deleted high voltage in the revised manuscript: “This work demonstrates the possibility of high ionic conductivity sodium ion halide SSEs, which would promote the development of sodium ASSBs.”

Referee Query 4:

I do not think the EXAFS data is reliable if the halide has hydrolyzed, as that degradation will fundamentally alter its properties. Air-sensitive sample transfer is a must.

Author response 4: We thank the reviewer for this valuable suggestion. The EXAFS was repeated with covering samples with air tight Kapton film and the results show no hydrolyzation. The following discussion was revised as below: “The Zr K-edge and La L₃-edge X-ray absorption fine structure (XAFS) was obtained to reveal the local coordination structure of Na_{0.7}La_{0.7}Zr_{0.3}Cl₄-HT, Na_{0.7}La_{0.7}Zr_{0.3}Cl₄-HM and NaLaCl₄-HT. It can be seen that the absorption edge of the Zr K-edge and La L₃-edge of three were basically coincide (Supplementary Fig. 14), indicating equal valence with Zr⁴⁺ and La³⁺. From the R-space of Zr K-edge, the main peak at about 2 Å could be recognized as Zr-Cl coordination from Na_{0.7}La_{0.7}Zr_{0.3}Cl₄-HT and Na_{0.7}La_{0.7}Zr_{0.3}Cl₄-HM (Supplementary Fig. 15). Similarly, the R-space curve of La L₃-edge also show a significant difference in the position of the main peaks compare with La₂O₃ (Supplementary Fig. 16), which could be recognized as La-Cl coordination of NaLaCl₄, Na_{0.7}La_{0.7}Zr_{0.3}Cl₄-HT and Na_{0.7}La_{0.7}Zr_{0.3}Cl₄-HM. As a result, Zr-Cl bond and La-Cl bond lengths are different obviously.”

“As shown in **Fig. 4**, EXAFS fitting is performed to quantitatively compare the Zr-Cl and La-Cl coordination structures in Na_{0.7}La_{0.7}Zr_{0.3}Cl₄ and the results are summarized in **Supplementary Tab. 12**. NaLaCl₄, NLZC0.3-HT and NLZC0.3-HM have similar bond length of La-Cl and the coordination numbers of La, which proves that the doping of the Zr element and ball milling process does not change its structure, which is consistent with the XRD results (Supplementary Fig. 1). It can be confirmed that the bond length of Zr-Cl is about 2.48 Å while it is 2.94 Å for La-Cl.”

Fig. 4 R-space fitting curves of $\text{Na}_{0.7}\text{La}_{0.7}\text{Zr}_{0.3}\text{Cl}_4$

Supplementary Table 12 Fitting results from EXAFS

Sample	R-factor	Path	N	R[Å]	ΔE_0 [eV]	$\sigma^2 [10^{-3} \text{Å}^2]$
Zr foil	0.0007	Zr-Zr	6*	3.16	5.7	3.0
		Zr-Zr	6*	3.27		3.0
ZrCl ₄	0.0076	Zr-Cl	2*	2.31	-1.52	1.3
		Zr-Cl	2*	2.47		2.8
		Zr-Cl	2*	2.63		5.2
Zr-NLZC0.3-HT	0.0024	Zr-Cl	7.3	2.48	1.26	3.7
Zr-NLZC0.3-HM	0.0031	Zr-Cl	6.9	2.48	0.82	6.5
La ₂ O ₃	0.0084	La-O	6*	2.53	15.1	14.0
La-NLC	0.0029	La-Cl	8.2	2.93	17.5	15.9
La-NLZC0.3-HT	0.0021	La-Cl	8.9	2.94	17.7	15.0
La-NLZC0.3-HM	0.0032	La-Cl	8.8	2.94	18.2	15.5

Supplementary Figure 14 (a)La L₃-edge and (b)Zr K-edge XANES spectra for the Na_{0.7}La_{0.7}Zr_{0.3}Cl₄ and NaLaCl₄

Supplementary Figure 15 R-space of Zr K-edge XANES spectra for Na_{0.7}La_{0.7}Zr_{0.3}Cl₄

Supplementary Figure 16 R space of La L_3 -edge XANES spectra for $\text{Na}_{0.7}\text{La}_{0.7}\text{Zr}_{0.3}\text{Cl}_4$ and NaLaCl_4

Referee Query 5:

In addition, there are concerns about the novelty of this manuscript.

Author response 5: The referee raises a good question. The first point is this is a new type Na halide SSE, which is potentially to achieve even high ionic conductivity and low activation energy. Our further work shows that the ionic conductivity can reach $6.6 \times 10^{-4} \text{ S cm}^{-1}$ and the activation energy can be reduced to 0.19 eV at least, as shown in Extended Data Figure 1. We would figure out the reason and publish these results later.

Secondly, the ASSB with LaCl_3 based Na^+ ion SSE shows better rate performance than from the literature (<https://www.nature.com/articles/s41467-021-21488-7>) and the preprint (<https://chemrxiv.org/engage/chemrxiv/article-details/637a89cf20798134fe2e6586>), after replacing $\text{Na}_{2.9}\text{PS}_{3.9}\text{Cl}_{0.1}$ with Na_3PS_4 as the transition layer in the ASSB. The changes are detailed in the Response Letter and are highlighted in blue font in the revised main text and figure captions:

Extended Data Figure 1 LaCl₃ based Na⁺ ion SSE with higher ionic conductivity and lower activation energy.

Fig. 5 (a) Configuration of ASSB with NLZC0.3-HM as SSEs, (b) Cross section SEM for different layers in the ASSB: (b1) NaCrO₂ composite cathode, (b2) Na_{0.7}La_{0.7}Zr_{0.3}Cl₄, (b3) Na₃PS₄, (b4)

Na₂Sn anode, (c) charge-discharge profiles at different currents (30 °C), (d) rate performance, (e) charge-discharge profiles at different cycles (30 °C, 0.3C), (f) cycle performance.

“When cycling at 0.1 C (30 °C), the discharge capacity for the first cycle reaches 123 mA h g⁻¹, with a high columbic efficiency (CE) of 95%, reflecting a highly irreversible reaction in this ASSB. When current density increases, the capacity decreases from 123 mA h g⁻¹ (0.1C) to 119 mA h g⁻¹ (0.2C), 114 mA h g⁻¹ (0.3C), 108 mA h g⁻¹ (0.5C), and 94 mA h g⁻¹ (1C), showing high-capacity retention at high current density (**Fig. 5c, d**). Compared to other halide SSEs with lower ionic conductivity, the rate performance of ASSB using NLZC0.3-HM is better.^{4,46}”

- A previous study has already elucidated the effect of Zr⁴⁺ aliovalent doping into sodium halide solid-state electrolytes. <https://www.nature.com/articles/s41467-021-21488-7>. The conclusions in this paper are very similar.

Author response: Aliovalent doping is quite common in enhancing ionic conductivity of SSEs. Zr⁴⁺ doping has a different affect in the LaCl₃-based SSE, which is changing the distributions of Na⁺ ions. More importantly, Zr⁴⁺ doping is the tool to enhance the ionic conductivity but not the key point in this work.

- There is a preprint from 2022 which has shown higher Na conductivity and longer cycling stability than the findings in this manuscript. <https://chemrxiv.org/engage/chemrxiv/article-details/637a89cf20798134fe2e6586>

Author response: Thanks for sharing this work, which we didn't notice before. We have cited this work and compared the rate performance in the revised manuscript.

REVIWER 2:

Referee comment:

In the manuscript "New Sodium Halide Solid Electrolytes with High Ionic Conductivity for All-Solid-State Batteries", the authors reported a new LaCl₃-based solid electrolyte with ionic

conductivity of 2.9×10^{-4} S/cm. When combined with Na₂Sn anode and NaCrO₂ cathode, the solid-state cell gives an initial capacity of 119 mA h/g at 0.1C. There are quite a few aspects that should be clarified in the manuscript. Some suggestions to further improve the paper:

Author response: We thank the reviewer for the detailed suggestions, which helps to improve the quality of our work.

Referee Query 1:

The authors mentioned that “As a result, the bond length of Na-Cl in the 1D channel increases from the original 2.87 Å to 2.91 Å (Fig. 3c), implying that the migration of Na⁺ ions requiring a lower barrier comparing to NLC (0.04 eV in NLC vs. 0.26 eV in NLZC, Fig. 3d).”, which may be not accurate. More explanations about the difference between NLZC and NLZC-La/Na mixing should be given. In actual situations, whether La/Na mixing occurs? If occurs, giving the simulated result about NLZC is reasonable or not?

Author response 1: Thank you very much for the reviewer's correction of this point of view. Our explanation about the relationship between Na-Cl bond length and activation energy is too simple, makes this not convincing. We added more explanations in the revised manuscript: “The longer Na-Cl bond broadens the diffusion bottleneck in the 1D channel, lowers the site energy of Na⁺ ions at the bottleneck, thereby lowering their migration activation energy and increasing the Na⁺ ion conductivity. The AIMD simulations imply that the migration of Na⁺ ions in NLZC requiring a lower barrier comparing to NLC (0.04 eV in NLC vs. 0.26 eV in NLZC, Fig. 3d).”

The La/Na mixing was confirmed with XRD refinement. More importantly, this is reasonable since La³⁺ (1.032 Å) has a similar size to Na⁺ (1.02 Å) and this La³⁺/Na⁺ mixing at 2c site is confirmed already (*Z. Anorg. Allg. Chem.* **620**, 444–450 (1994)). The blocking of 1D ion conduction is also found in other materials, such as LiFePO₄ (*Chem. Mater.* **17**, 5085–5092 (2005)), so this La³⁺/Na⁺ mixing should be considered when conducting theoretical simulations.

Energy is considered when introducing La³⁺ at 2b site. The model with the lowest energy was selected to calculate Na⁺ ion conductivity. Meanwhile, the thermodynamic stability of this optimized model was also investigated. Ion mixing generally increases the entropy in solids, making it relatively stable at high temperatures. AIMD results show that the mixed material can still maintain stability at 500K, which is one of the reasons for the mixing of materials obtained in

the experiment. More explanations are added in the revised manuscript: “The NLZC-La/Na-mixing structure was constructed with energy as the criterion (Supplementary Fig. 11). The thermodynamic stability of the constructed model was also investigated (Supplementary Fig. 12). The comparison of results shows that when the NLZC is at 400K, the structural frame with LaCl_3 as the premise has greater distortion, which means it is more unstable, while the NLZC-La/Na-mixing still keeps the structure relatively stable under 500K, indicating that the mixing makes the material more stable.”

Referee Query 2:

In Table S6 2b site, the occupancy of Na and La is 0.391 and 0.071. But the authors mentioned that “A mixing ratio (0.06) of La/Na at 2b site along the c axis was found from XRD refinement, due to the similar radius of La^{3+} and Na^+ (Table S6).” How can the authors give the numbers?

Author response 2: We are sorry for this mistake. We should be referring to **Table S8**, which is the sample ($\text{Na}_{0.7}\text{La}_{0.7}\text{Zr}_{0.3}\text{Cl}_4\text{-HM}$) with the highest ionic conductivity. The mixing ration here means the occupancy of La at 2b site, which is 0.06. To avoid misunderstanding, we revised this part in the revised manuscript “A small amount of La^{3+} (0.06) occupy the 2b site along the c axis, due to the similar radius of La^{3+} (1.032 Å) and Na^+ (1.02 Å) (Supplementary Tab. 8). The high valence of La^{3+} enhances the La-Cl bond energy and decreases its mobility, thus the La^{3+} on the migration channel should block the migration of Na^+ . Similar phenomena have also been found in other one-dimensional conductors, such as LiFePO_4 .⁴⁵”

Referee Query 3:

In absorption edge of the Zr K-edge, the comparison of ZrCl_4 samples should be considered. The comparison of ZrO_2 samples to prove the presence of Zr-Cl coordination is not convincing enough. The XAFS test is capable of avoiding hydrolysis. The fitting results of Zr-O and La-O should be excluded.

Author response 3: We thank the reviewer for this insightful suggestion. We repeated the EXAFS without the disruption of moist air. The following discussion was revised as below: “The Zr K-

edge and La L₃-edge X-ray absorption fine structure (XAFS) was obtained to reveal the local coordination structure of Na_{0.7}La_{0.7}Zr_{0.3}Cl₄-HT, Na_{0.7}La_{0.7}Zr_{0.3}Cl₄-HM and NaLaCl₄-HT. It can be seen that the absorption edge of the Zr K-edge and La L₃-edge of three were basically coincide (Supplementary Fig. 14), indicating equal valence with Zr⁴⁺ and La³⁺. From the R-space of Zr K-edge, the main peak at about 2 Å could be recognized as Zr-Cl coordination from Na_{0.7}La_{0.7}Zr_{0.3}Cl₄-HT and Na_{0.7}La_{0.7}Zr_{0.3}Cl₄-HM (Supplementary Fig. 15). Similarly, the R-space curve of La L₃-edge also show a significant difference in the position of the main peaks compare with La₂O₃ (Supplementary Fig. 16), which could be recognized as La-Cl coordination of NaLaCl₄, Na_{0.7}La_{0.7}Zr_{0.3}Cl₄-HT and Na_{0.7}La_{0.7}Zr_{0.3}Cl₄-HM. As a result, Zr-Cl bond and La-Cl bond lengths are different obviously.”

“As shown in **Fig. 4**, EXAFS fitting is performed to quantitatively compare the Zr-Cl and La-Cl coordination structures in Na_{0.7}La_{0.7}Zr_{0.3}Cl₄ and the results are summarized in **Supplementary Tab. 12**. NaLaCl₄, NLZC0.3-HT and NLZC0.3-HM have similar bond length of La-Cl and the coordination numbers of La, which proves that the doping of the Zr element and ball milling process does not change its structure, which is consistent with the XRD results (Supplementary Fig. 1). It can be confirmed that the bond length of Zr-Cl is about 2.48 Å while it is 2.94 Å for La-Cl.”

Fig. 4 R-space fitting curves of Na_{0.7}La_{0.7}Zr_{0.3}Cl₄

Supplementary Table 12 Fitting results from EXAFS

Sample	R-factor	Path	N	R[Å]	ΔE_0 [eV]	$\sigma^2 [10^{-3} \text{Å}^2]$
Zr foil	0.0007	Zr-Zr	6*	3.16	5.7	3.0
		Zr-Zr	6*	3.27		3.0
ZrCl ₄	0.0076	Zr-Cl	2*	2.31	-1.52	1.3
		Zr-Cl	2*	2.47		2.8
		Zr-Cl	2*	2.63		5.2
Zr-NLZC0.3-HT	0.0024	Zr-Cl	7.3	2.48	1.26	3.7
Zr-NLZC0.3-HM	0.0031	Zr-Cl	6.9	2.48	0.82	6.5
La ₂ O ₃	0.0084	La-O	6*	2.53	15.1	14.0
La-NLC	0.0029	La-Cl	8.2	2.93	17.5	15.9
La-NLZC0.3-HT	0.0021	La-Cl	8.9	2.94	17.7	15.0
La-NLZC0.3-HM	0.0032	La-Cl	8.8	2.94	18.2	15.5

Supplementary Figure 14 (a)La L₃-edge and (b)Zr K-edge XANES spectra for the Na_{0.7}La_{0.7}Zr_{0.3}Cl₄ and NaLaCl₄

Supplementary Figure 15 R-space of Zr K-edge XANES spectra for Na_{0.7}La_{0.7}Zr_{0.3}Cl₄

Supplementary Figure 16 R space of La L₃-edge XANES spectra for Na_{0.7}La_{0.7}Zr_{0.3}Cl₄ and NaLaCl₄

Referee Query 4:

In Supplementary Figure 12, how to get the oxidation potential of 3.7 V? It seems a clear deviation.

Author response 4: We thank the referee for the insightful question. Linear scanning voltammetry (LSV) was used to test the electrochemical window of the NLZC0.3-HM in order to better evaluate the electrochemical window. We have added the following discussion in revised manuscript: “Linear scanning voltammetry (LSV) of NLZC0.3-HM-Super P||NLZC0.3-HM||Na₃PS₄||Na₂Sn reveals that the electrochemical stable window of NLZC0.3-HM is from 1.33 V to 3.80 V vs. Na₂Sn at 30 °C and from 1.44 V to 3.79 V vs. Na₂Sn at 60 °C (Supplementary Fig. 17), which is consistent with the theoretical results of sodium halide SSEs.⁴”

Supplementary Figure 17 Linear scanning voltammetry of NLZC0.3-HM at 0.1 mV s⁻¹ and (a) 30°C, and (b) 60°C

Referee Query 5:

In cell tests, why test under 60 °C? Is it reasonable to test at high temperature when SE conductivity is high enough? Typically, lithium cells could test under room temperature with such ionic conductivity of SE.

Author response 5: We thank the referee for the insightful question. In the revised manuscript, we use Na_3PS_4 instead of $\text{Na}_{2.9}\text{PS}_{3.9}\text{Cl}_{0.1}$ as a barrier layer, which significantly reduces the interfacial resistance between NLZC0.3-HM and the barrier layer. This new configuration of ASSB can be operated at room temperature and shows good rate performance. The changes are detailed in the Response Letter and are highlighted in blue font in the revised main text and figure captions: “The ASSB of $\text{NaCrO}_2\|\text{NLZC0.3-HM}\|\text{Na}_3\text{PS}_4\|\text{Na}_2\text{Sn}$ was cycled between 2.0 V and 3.4 V, in which range the NLZC0.3-HM SSE is electrochemically stable. When cycling at 0.1 C (30 °C), the discharge capacity for the first cycle reaches 123 mA h g^{-1} , with a high columbic efficiency (CE) of 95%, reflecting a highly irreversible reaction in this ASSB. When current density increases, the capacity decreases from 123 mA h g^{-1} (0.1C) to 119 mA h g^{-1} (0.2C), 114 mA h g^{-1} (0.3C), 108 mA h g^{-1} (0.5C), and 94 mA h g^{-1} (1C), showing high-capacity retention at high current density (Fig. 5c, d). Compared to other halide SSEs with lower ionic conductivity, the rate performance of ASSB using NLZC0.3-HM is better.^{4,46} After cycling at 0.3 C for 70 times, the capacity retains 100 mA h g^{-1} , which is 88% of the initial capacity (Fig. 5e, f). Part of the capacity loss should come from an increase in resistance, which is 780Ω after 70 cycles at 0.3C (Supplementary Fig. 26). This increase in resistance may originate from $\text{Na}_3\text{PS}_4/\text{Na}_2\text{Sn}$ interface, further enhancement in cycle performance can be achieved with other SSE transition layers, such as $\text{Na}_4(\text{B}_{12}\text{H}_{12})(\text{B}_{10}\text{H}_{10})$.⁴⁶”

Referee Query 6:

The stability of the new SE against Na_2Sn is not good. However, the LaCl_3 -based Lithium SE is stable with Li metal anode. The authors need give more insightful investigation.

Author response 6: We thank the referee for the insightful question. The LaCl_3 -based Lithium SE is actually dynamically stable with Li metal anode, which means a stable interface layer forming at the SE/Li metal interface. This interface layer should ionic conductive but electronic insulate. When changed from Li to Na, the high reducibility of Na makes it difficult to form stable interface layer, even Na-Sn alloy cannot be dynamically stable with LaCl_3 -based Sodium SE. The $\text{Na}_2\text{Sn}\|\text{NLZC0.3-HM}\|\text{Na}_2\text{Sn}$ symmetric cell was cycled to investigate the stability of NLZC0.3-HM against Na_2Sn . Relevant results were added: “The high reduction potential of NLZC0.3-HM indicates that this SSE is thermodynamically unstable against Na_2Sn or Na metal anodes. To verify

the dynamic stability of NLZC0.3-HM against low potential anodes and its possibility as an anolyte, symmetric cell of $\text{Na}_2\text{Sn}||\text{NLZC0.3-HM}||\text{Na}_2\text{Sn}$ was assembled and cycled (Supplementary Fig. 18). When cycled at 0.01 mA cm^{-2} , the voltage increases to 0.1 V within only 70 h, showing the instability of $\text{Na}_2\text{Sn}/\text{NLZC0.3-HM}$ interface (Supplementary Fig. 18) and this SSE cannot be directly used as an anolyte. Both La 3d and Zr 3d peaks shift to the low energy direction, indicating the reduction of Zr^{4+} and La^{3+} after cycling (Supplementary Fig. 19).”

Supplementary Figure 18 (a) Galvanostatic cycling of the $\text{Na}_2\text{Sn}||\text{Na}_2\text{Sn}$ symmetric cells at 30°C .

(b) Nyquist plots of $\text{Na}_2\text{Sn}||\text{NLZC0.3-HM}||\text{Na}_2\text{Sn}$ before and after cycling.

Supplementary Figure 19 (a) La 3d and (b) Zr 3d X-ray photoelectron spectroscopy (XPS) spectra of NLZC0.3, and NLZC0.3-HM+ Na_2Sn .

REVIEWER 3:

Referee comment:

In this manuscript, the authors report on $\text{Na}_{1-x}\text{Zr}_x\text{La}_{1-x}\text{Cl}_4$ prepared by annealing and subsequent mechanochemical milling method as a halide solid electrolyte material with 0.29 mS cm^{-1} for all-solid-state Na^+ batteries. The structure of $\text{Na}_{1-x}\text{Zr}_x\text{La}_{1-x}\text{Cl}_4$ was characterized using XRD Rietveld refinement and EXAFS. It is interesting that a report about Na halide electrolyte with hexagonal structure is the first time. Besides, the electrochemical performance of $\text{Na}_{1-x}\text{Zr}_x\text{La}_{1-x}\text{Cl}_4$ was also characterized by cyclic voltammetry and half-cell tests. Including above issues, the following discussion points should be addressed extensively.

Author response 1: We are sincerely grateful to the referee for their highly constructive and positive review, which helps greatly improving the quality of the manuscript.

Referee Query 1:

Regarding the composition of ' $\text{Na}_{0.7}\text{Zr}_{0.3}\text{La}_{0.7}\text{Cl}_4$ ': Given that ZrCl_4 has melting and boiling points of $437 \text{ }^\circ\text{C}$ and $331 \text{ }^\circ\text{C}$, respectively, the annealing protocol, which is performed at $450 \text{ }^\circ\text{C}$ for 10 h in an Ar furnace, may pose a possibility of ZrCl_4 precursor sublimation prior to the intended reaction, potentially altering the nominal compositions?

Author response 1: We thank the reviewer for this important consideration. We double checked the possible sublimation of ZrCl_4 during the synthesis process and found limited sublimation product on the left of the tube, which is outside of the furnace during sintering and is with low temperature (Extended Data Figure 2).

Extended Data Figure 2 Optical photographs of NLZCx after sintering

The reason is that the high energy ball milling can not only fully mix raw materials, but also promote reactions between them. So, after ball milling and before sintering, $ZrCl_4$ has reacted with other materials to form other components with high boiling point. (Extended Data Figure 3).

Extended Data Figure 3 XRD patterns of NLZC0.4 before annealing

Referee Query 2:

The pivotal role of mechanochemical milling, which enhances conductivity by nearly two orders of magnitude, is broadly, limitedly, and vaguely discussed. Specifically, the claim “To further promote the ionic conductivity, the high temperature sintered samples were ball milled to introduce disordering and defects, thus rising the doping concentration of Zr^{4+} and further expanding the lattice and diffusion channel (NLZC_x-HM).” Warrants a meticulous discussion. Considering that the reported material has rapid 1D Na ion conduction channels along the c axis, introducing disorder might adversely impact the 1D connectivity, which necessitates an extensive and in-depth exploration supplemented by additional data and/or theories.

Author response 2: That is a really good question. Normally, ball milling can introduce defects, distortions etc., leading to changes in local structure. The structure change can affect the energy

and distribution of Na ions at different sites. As a result, the ion migration activation energy and ionic conductivity can be changed. The structure change with ball milling was further investigated with XAFS and synchrotron XRD. According to the synchrotron XRD, ball milling increased the number of Na⁺ ions at 2*b* site and also the lattice parameter *a*, which increases the charge carrier concentration and size of diffusion bottleneck, both are benefit to the ion migration. However, Ball milling did not excessively increase the disorder of the electrolyte, and the LaCl₃ framework structure was well maintained. Therefore, the adverse effect on conductivity is relatively small. Relevant figures and contents are added: “Synchrotron XRD of NaLaCl₄-HT and NaLaCl₄-HM were carried out to figure out the changes in fine structure and the reason for volume expansion. The refinement results shown in Supplementary Fig. 6 and Supplementary Tab. 9 reveal a small amount of Na⁺ at the 2*b* site (0.16) in NaLaCl₄-HT, which means this site is small and metastable for Na⁺ occupying. Excess Na exists in the form of unknown impurities (Supplementary Fig. 6). During the ball milling process, the high energy causes more Na⁺ to occupy the 2*b* site (0.251), resulting in disappearance of impurities and lattice expansion (Supplementary Fig. 7 and Supplementary Tab. 10). On the contrary, the expansion of the lattice makes the Na⁺ at the 2*b* site more stable. Zr⁴⁺ doping at La³⁺ site can also stabilize Na⁺ at 2*b* site and more Na⁺ occupies 2*b* site (0.461) in Na_{0.7}La_{0.7}Zr_{0.3}Cl₄-HT (Supplementary Fig. 8 and Supplementary Tab. 11), leading to an increase of lattice parameter *a* (7.583 Å). Furthermore, there is no significant change in the framework, such as significant decrease in orderliness or even to totally amorphous (Supplementary Fig. 6 and 7). This may be related to the strong bond energy of M-Cl, which keeps its main structure unchanged.”

Supplementary Figure 6 Synchrotron XRD refinements of NaLaCl₄-HT

Supplementary Figure 7 Synchrotron XRD refinements of NaLaCl₄-HM

Supplementary Figure 8 Synchrotron XRD refinements of Na_{0.7}La_{0.7}Zr_{0.3}Cl₄-HT

Referee Query 3:

Why does the lattice parameter expand upon although Zr is introduced in the lattice and Na vacancy is formed? The ionic radius of Zr⁴⁺ is smaller than that of La³⁺.

Author response 3: Many thanks to the reviewer for this constructive comment. The lattice expansion should originate from the increase of Na⁺ ions at 2b site. As Zr⁴⁺ replaces La³⁺ at the 2c

site, the shorter Zr-Cl bond causes elongation of Na-Cl bond at 2b site. The longer Na-Cl bond at 2b site makes this site more stable for Na⁺, while more Na⁺ ions occupying the 2b site further expands the lattice. Relevant figures and contents are added: “Zr⁴⁺ doping at La³⁺ site can also stabilize Na⁺ at 2b site and more Na⁺ occupies 2b site (0.461) in Na_{0.7}La_{0.7}Zr_{0.3}Cl₄-HT (Supplementary Fig. 8 and Supplementary Tab. 11), leading to an increase of lattice parameter a (7.583 Å).”

Supplementary Figure 8 Synchrotron XRD refinements of Na_{0.7}La_{0.7}Zr_{0.3}Cl₄-HT

Referee Query 4:

The electrochemical evaluation of NZLC should be supplemented, as the current state does not show any advantages of high ionic conductivity in NZLC. A comparison of rate performance with alternatives like Na_{2.25}Y_{0.25}Zr_{0.75}Cl₆ could be helpful.

Author response 4: Thank you very much for the reviewer's good suggestion. We have optimized the battery performance and tested the rate performance, which is better than Na_{2.25}Y_{0.25}Zr_{0.75}Cl₆. The changes are detailed in the Response Letter and are highlighted in blue font in the revised main text and figure captions: “The ASSB of NaCrO₂||NLZC0.3-HM||Na₃PS₄||Na₂Sn was cycled between 2.0 V and 3.4 V, in which range the NLZC0.3-HM SSE is electrochemically stable. When cycling at 0.1 C (30 °C), the discharge capacity for the first cycle reaches 123 mA h g⁻¹, with a high columbic efficiency (CE) of 95%, reflecting a highly irreversible reaction in this ASSB. When

current density increases, the capacity decreases from 123 mA h g⁻¹ (0.1C) to 119 mA h g⁻¹ (0.2C), 114 mA h g⁻¹ (0.3C), 108 mA h g⁻¹ (0.5C), and 94 mA h g⁻¹ (1C), showing high-capacity retention at high current density (**Fig. 5c, d**). Compared to other halide SSEs with lower ionic conductivity, the rate performance of ASSB using NLZC0.3-HM is better.^{4,46} After cycling at 0.3 C for 70 times, the capacity retains 100 mA h g⁻¹, which is 88% of the initial capacity (**Fig. 5e, f**). Part of the capacity loss should come from an increase in resistance, which is 780 Ω after 70 cycles at 0.3C (Supplementary Fig. 26). This increase in resistance may originate from Na₃PS₄/Na₂Sn interface, further enhancement in cycle performance can be achieved with other SSE transition layers, such as Na₄(B₁₂H₁₂)(B₁₀H₁₀).⁴⁶

Referee Query 5:

Cyclic voltammetry for oxidative and reductive stability should be measured separately at room temperature and 60 °C.

Author response 5: We are sincerely grateful to the referee for this suggestion. In order to display the electrochemical stability window of NLZC more clearly, linear scanning voltammetry (LSV) was conducted. We have added the following discussion in revised manuscript: “Linear scanning voltammetry (LSV) of NLZC0.3-HM-Super P||NLZC0.3-HM||Na₃PS₄||Na₂Sn reveals that the electrochemical stable window of NLZC0.3-HM is from 1.33 V to 3.80 V vs. Na₂Sn at 30 °C and from 1.44 V to 3.79 V vs. Na₂Sn at 60 °C (Supplementary Fig. 17), which is consistent with the theoretical results of sodium halide SSEs.⁴”

Supplementary Figure 17 Linear scanning voltammetry of NLZC0.3-HM at 0.1 mV s⁻¹ and (a)30°C, and (b) 60°C

Referee Query 6:

In line 221, “NLZC0.3-HN-super P||NLZC0.3-HM||Na_{2.9}PS_{3.9}Cl_{0.1}||Na” should be revised to “NLZC0.3-HN-super P||NLZC0.3-HM||Na_{2.9}PS_{3.9}Cl_{0.1}||Na₂Sn”.

Author response 6: We thank the reviewer for the correction. We have revised this as follows: “Linear scanning voltammetry (LSV) of NLZC0.3-HM-Super P||NLZC0.3-HM||Na₃PS₄||Na₂Sn reveals that the electrochemical stable window of NLZC0.3-HM is from 1.33 V to 3.80 V vs. Na₂Sn at 30 °C and from 1.44 V to 3.79 V vs. Na₂Sn at 60 °C (Supplementary Fig. 17), which is consistent with the theoretical results of sodium halide SSEs.⁴”

Referee Query 7:

The statement, “Since 2018, plenty of lithium-ion halide SSEs with high ionic conductivity up to 10-2 S/cm have been investigated, such as Li₃YCl₆, Li₃InCl₆, Li₃ScCl₆, Li₂ZrCl₆, and LiTaOCl₄ et al.29–34” is lacking original references detailing Li₂ZrCl₆: *Adv. Energy Mater.* 2021, 11, 2003190 and Li₃ScCl₆ (Li₂Sc_{2/3}Cl₄): *Energy Environ. Sci.* 2020, 13, 2056. Also, strictly, reference 34, concerning Li₃TiCl₆, cannot be categorized as a source pertaining to halide electrolytes.

Author response 7: We are sincerely grateful to the referee for sharing these papers. These references are added in the revised manuscript and ref. 34 is removed.

“34. Kwak, H. *et al.* New Cost-Effective Halide Solid Electrolytes for All-Solid-State Batteries: Mechanochemically Prepared Fe³⁺-Substituted Li₂ZrCl₆. *Adv. Energy Mater.* **11**, 2003190 (2021).
35. Zhou, L. *et al.* A new halospinel superionic conductor for high-voltage all solid state lithium batteries. *Energy Environ. Sci.* **13**, 2056–2063 (2020).”

Referee Query 8:

Figure S21: Experimental details, like whether the “NLZC+NPSC” cell indicates a simple Li-ion blocking symmetric cell with a bilayer or composite pellet, should be explicitly provided. Also, it

is important to carefully examine and discuss if any interfacial resistance exist between different electrolytes, especially at 60 °C, supported by appropriate citations.

Author response 8: We are sincerely grateful to the referee for this good suggestion. The “NLZC+NPSC” cell indicates a bilayer. We revised the description and replaced NPSC with NPS (NPS || Na₂Sn and NLZC || NPS || Na₂Sn, Supplementary Figure 28). The resistance between NPSC and NLZC did exist and keeps increasing with time, which is the main reason for the poor electrochemical performance. Also, X-ray photoelectron spectroscopy (XPS) shows obvious change of NPSC after contacting with NLZC after 5 days.

In the revised manuscript, Na₃PS₄ was chosen as the barrier layer and the electrochemical performance is significantly enhanced. It is reported that the interface between Na-Sn alloy and Na₃PS₄ is not completely stable, leading to a certain capacity loss in the ASSB (doi:10.26434/chemrxiv-2022-x7llq). The changes are detailed in the Response Letter and are highlighted in blue font in the revised main text and figure captions: “The total resistance of these ASSBs before cycling is about 400 Ω (Supplementary Fig. 26), which was contributed by NLZC0.3-HM (~150 Ω, Supplementary Fig. 6d), Na₃PS₄ SSE (~80 Ω, Supplementary Fig. 27), Na₂Sn anode side (~20 Ω, Supplementary Fig. 27 and Supplementary Fig. 28a), resistance between NLZC0.3-HM and Na₃PS₄ (~50 Ω, Supplementary Fig. 28a, b), and interface from NaCrO₂ cathode side (~100 Ω, Supplementary Fig. 26 and Supplementary Fig. 28b). Na_{2.9}PS_{3.9}Cl_{0.1} with higher ionic conductivity (Supplementary Fig. 29) was also selected as a transition layer to reduce the total resistance of ASSB. However, this electrolyte reacts with NLZC0.3-HM and the resistance keeps increasing when co-pressed together (Supplementary Fig. 30). The XPS spectra indicate structural changes with NLZC0.3-HM and NPSC mixing together (Supplementary Fig. 31), especially for NPSC, which causes an increase in resistance (Supplementary Fig. 30).”

“The ASSB of NaCrO₂||NLZC0.3-HM||Na₃PS₄||Na₂Sn was cycled between 2.0 V and 3.4 V, in which range the NLZC0.3-HM SSE is electrochemically stable. When cycling at 0.1 C (30 °C), the discharge capacity for the first cycle reaches 123 mA h g⁻¹, with a high columbic efficiency (CE) of 95%, reflecting a highly irreversible reaction in this ASSB. When current density increases, the capacity decreases from 123 mA h g⁻¹ (0.1C) to 119 mA h g⁻¹ (0.2C), 114 mA h g⁻¹ (0.3C), 108 mA h g⁻¹ (0.5C), and 94 mA h g⁻¹ (1C), showing high-capacity retention at high current density (Fig. 5c, d). Compared to other halide SSEs with lower ionic conductivity, the rate performance

of ASSB using NLZC0.3-HM is better.^{4,46} After cycling at 0.3 C for 70 times, the capacity retains 100 mA h g⁻¹, which is 88% of the initial capacity (Fig. 5e, f). Part of the capacity loss should come from an increase in resistance, which is 780 Ω after 70 cycles at 0.3C (Supplementary Fig. 26). This increase in resistance may originate from Na₃PS₄/Na₂Sn interface, further enhancement in cycle performance can be achieved with other SSE transition layers, such as Na₄(B₁₂H₁₂)(B₁₀H₁₀).^{46,,}

Supplementary Figure 28 Nyquist plots of Na₂Sn and Na₃PS₄ two layers (a), Na₂Sn, Na₃PS₄ and NLZC0.3-HM three layers (b).

Supplementary Figure 30 Nyquist plots of NLZC0.3-HM and Na_{2.9}PS_{3.9}Cl_{0.1} double layer SSE.

Supplementary Figure 31 (a) Zr 3d, (b) S 2p, (c) P 3d and (d) La 3d X-ray photoelectron spectroscopy (XPS) spectra of NLZC0.3, $\text{Na}_{2.9}\text{PS}_{3.9}\text{Cl}_{0.1}$ and NLZC0.3-HM+NPSC.

REVIEWERS' COMMENTS

Reviewer #2 (Remarks to the Author):

The authors have appropriately addressed the most questions and comments. The manuscript can be accepted for the publication. But "(0.04 eV in NLC vs. 0.26 eV in NLZC, Fig. 3d)" is still wrong, and it should be "(0.04 eV in NLZC vs. 0.26 eV in NLC, Fig. 3d)".

Editorial note: Reviewer #2 was additionally asked to comment in the place of Reviewer #1. The additional comments are as follows:

I believe the authors have well addressed concerns from Review #1.

Reviewer #3 (Remarks to the Author):

Most of the issues raised by the reviewers have been addressed. The manuscript has been revised accordingly.

A remaining comment concerns the title, which is currently too broad and general. It must be more specific and include "LaCl₃-based sodium halide solid electrolytes"

Response Letter

Title: New Sodium Halide Solid Electrolytes with High Ionic Conductivity for All-Solid-State Batteries

REVIEWER 2:

Referee comment 1:

The authors have appropriately addressed the most questions and comments. The manuscript can be accepted for the publication. But “(0.04 eV in NLC vs. 0.26 eV in NLZC, Fig. 3d)” is still wrong, and it should be “(0.04 eV in NLZC vs. 0.26 eV in NLC, Fig. 3d)”

Author response 1: We are sorry about the misunderstanding. The following discussion was revised as below: “(0.04 eV in NLZC vs. 0.26 eV in NLC, Fig. 3d).”

REVIEWER 3:

Referee comment 1:

Most of the issues raised by the reviewers have been addressed. The manuscript has been revised accordingly. A remaining comment concerns the title, which is currently too broad and general. It must be more specific and include “LaCl₃-based sodium halide solid electrolytes.”

Author response 1: We thank the reviewer for this insightful suggestion. The title has been revised to “LaCl₃-Based Sodium Halide Solid Electrolytes with High Ionic Conductivity for All-Solid-State Batteries”.